# RSA: Recursive Sparse Attention with Hierarchical Deep–Shallow Memory and Sparse Activation

## Abstract

Linear sequence models offer strong efficiency advantages for long-context modeling due to their linear complexity. However, they still lag behind standard Transformers on many tasks, especially those requiring long-range reasoning and retrieval. Recent work has explored gating mechanisms, delta rules, and multi-states to narrow this gap. Despite these efforts, most existing methods primarily focus on short-range information, which limits their effectiveness on complex long-horizon tasks. Fundamentally, this limitation stems from insufficient information utilization and constrained effective memory capacity. Motivated by insights from neuroscience, we introduce a biologically inspired shallow–deep memory architecture, in which multiple memory states are connected and superposed in a structured manner. The shallow memory captures coarse-grained representations, while the deep memory stores residual information. We show that this design can theoretically match the storage capacity of standard attention. Furthermore, we adopt a sparse-attention-like readout mechanism that effectively enhances attention concentration. By jointly designing the memory storage and retrieval processes, we propose Recursive Sparse Attention (RSA), a novel attention mechanism that bridges the gap between linear models and standard attention. RSA establishes a principled foundation for linear architectures to approach, and potentially surpass, the expressive power of full attention. Empirically, language models built upon RSA achieve performance comparable to or exceeding that of Gated DeltaNet, RWKV-7, and Transformer across a diverse set of language benchmarks.

[1]Anonymous Institution, Anonymous City, Anonymous Region, Anonymous Country. Correspondence to: Anonymous Author <anon.email@domain.com>.

Preliminary work. Under review by the International Conference on Machine Learning (ICML). Do not distribute.

## 1. Introduction

Transformer (Vaswani et al., 2017) has become the dominant architecture for generative models represented by large language models (Touvron et al., 2023; Jiang et al., 2023; DeepSeek-AI et al., 2024; Dubey et al., 2024; Yang et al., 2025a). Its block-wise matrix operations (Dao, 2023) can fully utilize contemporary graphics processing units, and combined with autoregressive sequence modeling algorithms, it has significantly advanced the development of artificial intelligence. However, the core module—the standard Attention mechanism—requires exact storage of the entire sequence. This leads to two major inefficiencies in long-sequence tasks: the quadratic complexity of full-sequence connections during training and the unbounded growth of the KV Cache during inference. Moreover, when the sequence length is large, the fully-connected standard Attention introduces more noise compared to sparse connections, which under limited data and training resources can result in sub-optimal performance. To address these issues, numerous efficient and high-performance Attention variants have emerged, such as sparse Attention (Ge et al., 2024; Xiao et al., 2024), linear Attention (Katharopoulos et al., 2020; Qin et al., 2024; Yang et al., 2024), and alternative sequence modeling approaches like State Space Models (SSMs) (Dao & Gu, 2024).

**Sparse Attention** methods typically select a subset of the sequence for actual Attention computation. We broadly categorize them into static sparse and dynamic routing approaches. **Static methods**—e.g., Swin Transformer (Liu et al., 2021), BigBird (Zaheer et al., 2020), and Attention-sink (Xiao et al., 2024) preselect fixed positions such as local windows, shifted windows, global tokens, or random positions. While these improve efficiency, their fixed selection pattern may discard information outside the chosen positions, leading to unacceptable performance degradation in certain tasks. **Dynamic routing methods**, such as Routing Transformer (Roy et al., 2021), MoBA (Lu et al., 2025), and NSA (Yuan et al., 2025), retrieve the most relevant key-value pairs on-the-fly. NSA further incorporates a compression branch to capture global coarse information. Dynamic routing outperforms static approaches because it can recover the static selection pattern when that is optimal.

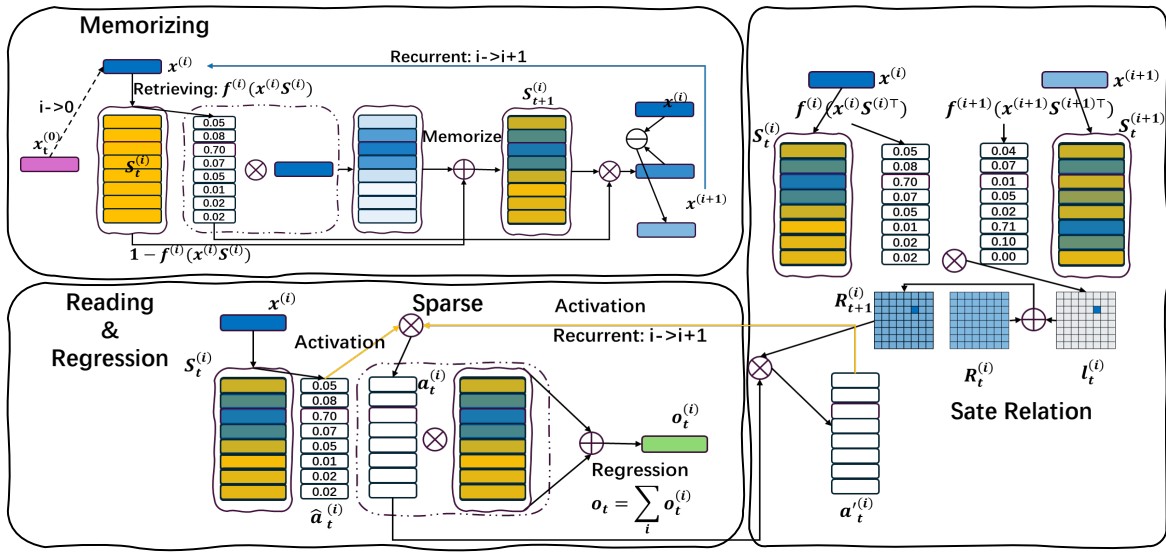

*Figure 1.* (a) **Deep–shallow memory storage.** At time step (t), the input is compressed into the shallow state, and the residual error with respect to this state is recursively passed to deeper states. This process mimics human memory, where shallow layers store coarse information and deeper layers store precise residual details. (b) **State-relation memory.** Relations between shallow and deep states are stored to encode their connectivity, enabling preliminary sparsification during retrieval. (c) **Memory readout.** The readout combines state-relation links with attention-based activation filtering to achieve focused coefficient retrieval. Here, $\otimes$ denotes matrix multiplication, and transposition is applied when appropriate.

However, its drawback is the extra computational overhead introduced by the routing process, which trades off with the actual Attention computation and prevents linear complexity. It also requires **storing the full fine-grained KV Cache**.

**Linear model** methods maintain a fixed-size state and enhance performance through various update mechanisms. Early Linear-Attention approaches replaced the softmax with a kernel function and rearranged computation order to obtain a fixed-size KV state. However, their over-dispersed attention limited performance. Subsequent works, exemplified by GLA (Yang et al., 2024), introduced gating mechanisms to sharpen the update, effectively focusing more on nearby tokens, akin to sliding-window Attention. Yet, due to dynamic decay, such methods struggle with long-sequence and retrieval tasks. Recently, Delta-rule-based linear models (Schlag et al., 2021; Behrouz et al., 2024; Yang et al., 2025b) have been developed; they remove redundant information during state updates, greatly improving state utilization. To further boost information capacity, multi-state designs have been explored, e.g., MVA (Wang et al., 2025b), MoM (Du et al., 2025), and PLA (Wang et al., 2025a). These efforts have substantially narrowed the capability gap between linear models and standard Attention, even surpassing it in some short-sequence tasks.

Building on existing theoretical error analyses, and inspired by neuroscience (Kveim et al., 2024), we propose a deep-shallow memory storage scheme coupled with a dynamically sparsified read-out mechanism, analogous to human reasoning processes. This design aims to provide foundational components and facilities for matching or even surpassing standard Attention. The deep-shallow memory mimics human memory: shallow memory stores coarse information for fast access to simple queries, while deep memory stores residual fine-grained information related to the shallow content. Dynamic routing retrieves and adds this residual to the shallow representation, forming precise information. We name this framework Recursive Sparse Attention with Deep-Shallow Memory States (RSA). RSA combines the strengths of sparse and linear Attention while mitigating their weaknesses, achieving linear time-space complexity and explicitly filtering out noise via dynamic sparse reading.

We train language models of varying scales using RSA. For commonsense reasoning, language modeling, few-shot complex tasks, retrieval, and long-sequence tasks, RSA matches or surpasses state-of-the-art methods such as MoM, Gated DeltaNet, Titans, and RWKV.

## 2. Related Work and Preliminaries

### 2.1. Standard Attention

The Transformer architecture relies on the attention mechanism to dynamically compute contextualized representations. Given an input sequence $X \in \mathbb{R}^{t \times d}$, it is linearly projected into queries $Q \in \mathbb{R}^{t \times d_k}$, keys $K \in \mathbb{R}^{t \times d_k}$, and values $V \in \mathbb{R}^{t \times d_v}$. Standard Attention (SA) is computed as

$$O = \mathrm{SA}(Q, K, V) = \mathrm{Softmax}\left(\frac{QK^\top}{\sqrt{d_k}} \odot M\right)V,$$

where $M$ denotes the causal mask $d_k$ is the feature dimension used for normalization.

For clarity and to facilitate comparison with linear attention models that update states sequentially, we also express standard attention in a temporal form. Let $q_t \in \mathbb{R}^{1 \times d_k}$, $k_t \in \mathbb{R}^{1 \times d_k}$, and $v_t \in \mathbb{R}^{1 \times d_v}$ denote the query, key, and value at time step $t$. The output at time $t$ is given by

$$o_t = \sum_{i=1}^{t} \frac{\exp\left(\frac{q_t k_i^\top}{\sqrt{d_k}}\right) v_i}{\sum_{j=1}^{t} \exp\left(\frac{q_t k_j^\top}{\sqrt{d_k}}\right)} = \frac{\exp\left(\frac{q_t K_t^\top}{\sqrt{d_k}}\right) V_t}{\exp\left(\frac{q_t K_t^\top}{\sqrt{d_k}}\right)},$$

where $K_t = k_{1:t} \in \mathbb{R}^{t \times d_k}$ denotes the complete sequence of keys from the beginning up to step $t$, and $V_t = v_{1:t}$ is defined analogously.

## 2.2. Sparse Attention: Fixed-Pattern vs. Dynamic Routing

Sparse attention methods reduce computational complexity by restricting the set of key positions accessible to each query. Depending on whether the selection strategy depends on the input content, sparse attention can be broadly categorized into *fixed-pattern sparse attention* and *dynamically routed sparse attention*.

In a general form, sparse attention can be written as

$$\tilde{K} = \text{Loc}(K, i_k), \tilde{V} = \text{Loc}(V, i_k), \text{Output} = \text{Softmax}(Q\tilde{K})\tilde{V},$$

where $\text{Loc}(\cdot)$ denotes a selection operator and $i_k$ represents the set of selected key indices.

In **fixed-pattern sparse attention**, the index set $i_k$ is independent of the actual values of $Q$ and $K$, and is instead determined by predefined rules. Representative examples include: a) Sliding-window attention, where $i_k \in (i - \frac{s}{2}, i + \frac{s}{2})$; b) Global tokens or attention sinks, which select the first few positions $i_k \in (0, s)$; c) Swin Transformer, which alternates between local window attention and shifted window attention. While these methods offer clear computational benefits, their fixed selection patterns may lead to information loss in scenarios involving long-range dependencies or unevenly distributed salient information.

**Dynamically routed sparse attention**, in contrast, determines the index set based on content-adaptive mechanisms, selecting keys according to the relevance between $Q$ and $K$:

$$\tilde{K} = \text{Loc}(K, i_k(Q, K)), \quad \tilde{V} = \text{Loc}(V, i_k(Q, K)).$$

This class of methods improves expressive power while maintaining sparsity. For example, MoBA and Bi-Former (Zhu et al., 2023) segment and compress $Q$ and $K$, and select top-$K$ key regions based on segment-level similarity:

$$i_k(Q, K) = \arg\max_{\text{top-}K} \left(\bar{Q}\bar{K}^\top\right),$$

where $\bar{Q}, \bar{K} \in \mathbb{R}^{n/s \times d}$ denote segment-level representations. By dynamically selecting salient regions, routing-based methods typically outperform fixed-pattern sparse attention in both flexibility and performance. NSA further introduces an additional compression branch. We note that an explicit sliding-window branch is not strictly necessary, as dynamic routing can converge to such patterns; in practice, this branch mainly facilitates efficient Triton (Tillet et al., 2019) or CUDA kernel implementations.

## 2.3. Linear and Recursive Attention Mechanisms

Linear attention avoids explicit computation of $QK^\top$ by approximating the softmax similarity function with kernel methods. Its general form can be written as

$$\text{Output} = \text{Sim}(Q, K)V = \phi(Q)\phi(K)^\top V,$$

where $\phi(\cdot)$ denotes a feature mapping. Common designs include Linear Attention with $\phi(x) = \text{elu}(x) + 1$. Linear attention can be equivalently expressed in a parallel form or a recursive form. In the recursive formulation, a fixed-size state is maintained:

$$S_t = S_{t-1} + \phi(k_t)^\top v_t, \quad o_t = \phi(q_t)S_t,$$

which enables constant memory usage during inference. However, due to the absence of softmax normalization and positional bias, linear attention methods commonly suffer from attention dilution.

To mitigate this issue, Gated Linear Attention (GLA) introduces an input-dependent gating decay mechanism:

$$S_t = \text{diag}(g_t)S_{t-1} + k_t^\top v_t,$$

where the gating vector $g_t$ provides dynamic forgetting, encouraging the model to focus on nearby tokens and improving stability in long-sequence modeling. However, this design also introduces a structural gap from softmax attention, limiting the inheritance of its weight structure.

Another line of work compresses keys and values online while retaining softmax computation at the output stage. For instance, Gated Slot Attention (GSA) recursively updates compressed key–value representations:

$$K_t = \text{diag}(g_t)K_{t-1} + (1 - g_t)^\top k_t, \qquad (1)$$
$$o_t = \text{Softmax}(q_t K_t^\top)V_t.$$

The calculation process for value is the same. Such methods reduce computational complexity while preserving the softmax weight structure, leading to improved convergence behavior. Finally, **Delta Rule** memory mechanisms introduce an error-driven correction term by explicitly querying the previous state before updating:

$$S_t = S_{t-1} + g_t \cdot k_t^\top \left(v_t - k_t S_{t-1}\right),$$

which significantly enhances state capacity and the ability to model long-term dependencies.

## 3. Method

We posit that existing efficient sequence models have not yet demonstrated a convincing ability to match the performance of the standard Transformer across intuitive, theoretical, and empirical dimensions. While each line of work offers unique advantages, the observed gaps remain significant. Our goal is to synthesize these strengths into an algorithm that is *intuitively aligned with*, *theoretically comparable to*, and *empirically competitive with* standard attention, while operating in linear time and space.

**Core Premise.** We identify two essential components of standard attention: (1) *precise, unbounded information storage*, and (2) *attention-concentrated readout for regression*. For a linear model to serve as a functional replacement, it must equivalently preserve all key-value information and retrieve it with focus rather than diffusion. Our central thesis is that performance parity can be achieved by *discriminatively storing all information* and subsequently *reading relevant content with concentrated attention*, leveraging abundant coarse context for auxiliary localization.

**Storage Intuition.** The finite precision of digital computation imposes a fundamental upper bound on the number of distinct key-value token types. This permits a fixed-size state to approximate the unbounded storage capacity of standard attention (Wang et al., 2025a). We later formalize this via information theory, showing that densely connected, multi-level states are required. We instantiate this as a *deep–shallow memory* mechanism.

**Readout Intuition.** Selecting only the most relevant key-value tokens for autoregression naturally filters noise, leading to more accurate prediction. However, discarding all "irrelevant" context is suboptimal; a compressed representation of this information can usefully model the influence of distant tokens. We therefore employ *correlation-based dynamic routing with multi-scale compressed noise* to achieve focused readout and a final convex combination over values.

In the following sections, we detail our two core components: the deep-shallow memory storage and a recursively refined, correlation-based sparse readout mechanism.

### 3.1. Deep–Shallow Memory

**Intuitive Explanation:** The deep–shallow memory mechanism is inspired by how humans store and retrieve information. Human memory often evolves from concrete to abstract, from coarse to precise, or from shallow to deep representations. For example, when memorizing concepts, humans typically categorize information based on preferences or high-level features. Coarse categories determined by fuzzy attributes are stored at levels that can be accessed quickly, enabling rapid approximate reasoning, such as distinguishing broad biological species. When dealing with

more complex tasks, finer and more precise memories are required. These precise memories are usually dependent on coarse ones, or equivalently, stored as residuals relative to shallow memory, which facilitates later reconstruction of accurate information. For instance, distinguishing individuals within the same species relies on subtle differences encoded as residual information. Even more complex reasoning, such as character traits across long time spans, may require increasingly fine-grained residual representations.

We design the deep–shallow memory module by combining information-theoretic arguments, KV quantization (Hooper et al., 2025), and the upper bound on KV token types established in PLA (Wang et al., 2025a), guided by the above neuroscientific intuition. Consider standard attention with 4-bit quantization. A key vector of dimension $d_k$ can represent at most $2^{4d_k}$ distinct types. When multiple identical keys occur, they can be counted rather than stored redundantly. Therefore, to match the memory capacity of standard attention, a multi-state system must achieve a comparable number of representable KV token types.

**Theorem 1.** Given $n$ key states, where the $i$-th state has size $m_i$, the total number of distinct KV token types representable by these states is $\prod_{i=1}^{n} m_i$.

This result has appeared implicitly in various forms across prior works (Wang et al., 2025b). To achieve the aforementioned high storage capacity, MVA recursively stores information into the state using $K_t^{(i+1)} = K_t^{(i)} -$ Softmax$(K^{(i)} W_C^{\top}) K_t^{(i)}$ and performs retrieval via learnable parameters $W_C$, rather than directly querying the stored states like PLA. In contrast, MoM adopts a formulation of the form $K_t^{(i)} = x_t^{(0)} W^{(i)}$ where the state is constructed through layer-wise linear projections of the input. Our goal is to unify these multi-state theories and provide a neuroscientific interpretation that establishes a principled foundation for further development.

**Theorem 2.** For $n$ key states with sizes $\{m_i\}_{i=1}^{n}$, let $S_t^{K^{(i)}}$ denote the $i$-th key state at time $t$, and let $k_t^{(i)}$ denote the corresponding key vector. Any computation capable of representing $\prod_{i=1}^{n} m_i$ token types can be approximated by the following recursive formulation:

$$k_t^{(i+1)} = \text{MLP}\Big( k_t^{(i)} - \text{SA}\Big( k_t^{(i)}, S_t^{K^{(i)}}, S_t^{K^{(i)}} \Big) \Big),$$

where $\text{SA}(\cdot)$ denotes standard attention. For proof, please refer to Appendix C.

Interestingly, this formulation is structurally equivalent to a single Transformer layer. We believe this connection offers insights into model interpretability. Each Transformer layer can be viewed as performing classification at a certain granularity: earlier layers extract coarse features or broad categories, while later layers refine these representations

by adding increasingly precise residual information. The final prediction head aggregates features from coarse to fine scales to perform classification or regression. From this perspective, many tasks—including autoregressive modeling—can be decomposed into progressively refined classification problems. This viewpoint may also help explain emergent abilities, which arise only after representations reach sufficient granularity, analogous to how simple physical laws at atomic scales give rise to complex macroscopic phenomena. We leave a deeper interpretability analysis to future work.

**State update implementations.** The state $S_t^{K(i)}$ can be implemented in multiple ways. For example, following GSA or MVA, one may use equation 1.

Alternatively, inspired by DeltaNet, we adopt a Delta-rule–like formulation to explicitly remove redundant information. We integrate these approaches and implement them efficiently using Triton kernels:

$$S_t^{K(i)} = S_{t-1}^{K}{}^{(i)} - \frac{1}{l_t^{(i)}}^\top \bar{f}^{(i)}(\cdot) S_{t-1}^{K}{}^{(i)} + \frac{1}{l_t^{(i)}} \cdot \bar{f}^{(i)}(\cdot)^\top k_t^{(i)},$$

$$\bar{f}^{(i)}(\cdot) = \bar{f}^{(i)}\left(k_t^{(i)} S_{t-1}^{K(i)\top}\right), l_t^{(i)} = l_{t-1}^{(i)} + \bar{f}^{(i)}\left(k_t^{(i)} S_{t-1}^{K(i)\top}\right)^\top$$

where $\bar{f}^{(i)}$ can be either the sigma function or the Softmax function, and $l_t^{(i)}$ tracks the total amount of keys aggregated into each state vector, enabling clustering and averaging of correlated keys. The key point of the aforementioned process is the current key vector queries the existing state, removes redundant components proportional to their correlation, and then adds the new key. If the key is fully correlated, it is entirely absorbed; if uncorrelated, its contribution vanishes asymptotically as $l_t^{(i)}$ grows. Consequently, correlated keys are clustered into the same state vector.

**Inter-layer connectivity.** Finally, we record the connections between adjacent layers:

$$R_t^{(i,i+1)} = \left(1 - \left(\frac{1}{l_t^{(i)}}\right)^\top \bar{f}^{(i)}\!\left(k_t^{(i)} S_{t-1}^{K(i)\top}\right)\right) R_{t-1}^{(i,i+1)} + \frac{1}{l_t^{(i)}} \cdot r_t^{(i,i+1)}$$

$$r_t^{(i,i+1)} = \bar{f}^{(i)}\left(k_t^{(i)} S_{t-1}^{K(i)\top}\right)^\top \bar{f}^{(i+1)}\left(k_t^{(i)} S_{t-1}^{K(i)\top}\right) \in \mathbb{R}^{m_i \times m_{i+1}}.$$

### 3.2. Correlation-Aware Sparse Readout

Most existing linear sequence models primarily focus on state storage and update mechanisms. Delta-rule-based updates and densely connected multi-state designs have become increasingly mature and powerful. However, the readout process and the subsequent aggregation over value sequences have received comparatively little attention. We argue that this component is equally critical. Motivated by insights from neuroscience and sparse attention theory, we propose a correlation-aware sparse readout mechanism.

Specifically, state retrieval is guided by token-wise relevance: highly correlated states are selectively retrieved, while weakly correlated information is compressed or filtered and retained only as auxiliary signals for later regression.

**Intuition.** When answering a query, humans retrieve and process relevant memories rather than exhaustively recalling all stored information. Standard attention mechanisms perform a dense readout: all keys are retrieved, normalized via softmax, and applied to all values through a convex combination. While this strategy is effective for short sequences or low-information tasks, it becomes suboptimal for long sequences with complex and noisy information.

In long-context scenarios, irrelevant or noisy tokens may dominate after normalization, significantly diluting the contribution of truly relevant information. For instance, in biological classification tasks, discriminative features of the organism are critical, whereas environmental context often serves only as weak auxiliary information. As the environmental scope grows, it may even interfere with classification. In extreme cases, when the object of interest occupies only a small fraction of the visual field, it may be entirely overlooked, leading to misclassification. Nevertheless, coarse environmental context may still provide useful prior information, motivating controlled compression rather than complete removal.

Furthermore, fine-grained classification (e.g., distinguishing dog breeds) requires retrieving residual features on top of coarse representations. Thus, it is natural to use shallow, coarse memory to localize relevant deep residual memory, and then combine them. This principle also extends to logical and mathematical reasoning, where coarse representations correspond to entities, while fine residuals encode abstract relations. Decoupling entities and relations enables deeper reasoning within a tighter representational bound. Overall, we posit that information processing should proceed by iteratively retrieving fine residuals using coarse relevance signals, while compressing unrelated information for auxiliary regression.

**Readout formulation.** Given a query $q_t$ at time step $t$, we first retrieve the shallow (level-0) state:

$$\hat{a}_t^{(0)} = \sigma\left(q_t S_t^{K(0)\top}\right) W^{(0)} \in \mathbb{R}^{1 \times m_0},$$

where $\sigma(\cdot)W^{(0)}$ replaces the conventional softmax projection and empirically yields better performance. The resulting $\hat{a}_t^{(0)}$ serves as a relevance score for shallow states.

To induce sparsity, instead of dense normalization, we apply a relevance-preserving compression:

$$a_t^{(0)} = \text{ReLU}\left(\hat{a}_t^{(0)} - \frac{1}{z_0}\right) + \frac{1}{z_0 m_0},$$

*Table 1.* Common-sense reasoning performance. Models marked with * are reported directly from prior work (Titans, Gated DeltaNet, RWKV-7, etc.). All other results are obtained from our own training and evaluation.

| Model | Wiki ppl↓ | LMB ppl↓ | LMB acc↑ | PIQA acc↑ | Hella. acc↑ | Wino. acc$_n$ ↑ | ARC-e acc↑ | ARC-c acc↑ | SIQA acc$_n$ ↑ | BoolQ acc↑ | Avg acc↑ | MMLU acc↑ |
|---|---|---|---|---|---|---|---|---|---|---|---|---|
| *360M parameters / 15B tokens* | | | | | | | | | | | | |
| Transformer++* | 31.52 | 41.08 | 30.76 | 62.98 | 34.76 | 50.53 | 45.21 | 24.05 | 36.81 | 58.24 | 42.92 | – |
| GLA* | 28.51 | 43.02 | 28.73 | 64.05 | 35.96 | 50.00 | 54.19 | 24.29 | 37.13 | 58.39 | 44.09 | – |
| Mamba* | 30.83 | 40.21 | 29.94 | 63.79 | 35.88 | 49.82 | 49.24 | 24.56 | 35.41 | 60.07 | 43.59 | – |
| DeltaNet* | 28.65 | 47.30 | 28.43 | 63.52 | 35.95 | 49.63 | 52.68 | 25.37 | **37.96** | 58.79 | 44.04 | – |
| TTT* | 27.44 | 34.19 | 30.06 | 63.97 | 35.71 | 50.08 | 53.01 | 26.11 | 37.32 | 59.83 | 44.51 | – |
| Gated DeltaNet* | 27.01 | 30.94 | 34.11 | 63.08 | 38.12 | 51.60 | 55.28 | 26.77 | 34.89 | 59.54 | 45.42 | – |
| Titans (LMM)* | 26.18 | 29.97 | **34.98** | 64.73 | 39.61 | 51.85 | 55.60 | 28.14 | 34.52 | 59.99 | 46.17 | – |
| Transformer-Qwen | 30.46 | 39.08 | 29.70 | 65.18 | 37.96 | 49.17 | 56.17 | 26.45 | 36.96 | 59.54 | 45.14 | 25.8 |
| Gated DeltaNet | 27.51 | 31.25 | 33.86 | 63.12 | 36.82 | 50.89 | 55.04 | 26.86 | 34.68 | 59.06 | 45.04 | 24.2 |
| RSA-360M-q (Ours) | **25.89** | **28.89** | 34.98 | **67.26** | **39.98** | **52.67** | **56.70** | 28.34 | 35.22 | 59.83 | **46.87** | 25.3 |
| RSA-360M (Ours) | 27.05 | 35.86 | 32.89 | 66.50 | 39.24 | 52.04 | 56.70 | 27.81 | 37.24 | **60.12** | 46.57 | **26.4** |
| *1.3B parameters / 100B tokens* | | | | | | | | | | | | |
| RetNet* | 19.08 | 17.27 | 40.52 | 70.07 | 49.16 | 54.14 | 67.34 | 33.78 | 40.78 | 60.39 | 52.02 | – |
| HGRN2* | 19.10 | 17.69 | 39.54 | 70.45 | 49.53 | 52.80 | 69.40 | 35.32 | 40.63 | 56.66 | 51.79 | – |
| Mamba2* | 16.56 | **12.56** | 45.66 | 71.87 | 55.67 | 55.24 | 72.47 | 37.88 | 40.20 | 60.13 | 54.89 | – |
| Gated DeltaNet* | 16.42 | 12.17 | 46.65 | 72.25 | 55.76 | 57.45 | 71.21 | 38.39 | 40.63 | 60.24 | 55.32 | – |
| Gated DeltaNet | 16.40 | 12.72 | 44.92 | 71.86 | 54.12 | 56.64 | 71.50 | 37.82 | 40.78 | 60.01 | 54.70 | 26.2 |
| RSA-1.3B-q (Ours) | **16.24** | 12.06 | 46.82 | 72.96 | 55.64 | **57.72** | 72.54 | **38.68** | 40.82 | 60.89 | **55.76** | 26.1 |
| RSA-1.3B (Ours) | 16.97 | 13.06 | 45.41 | 72.80 | **55.88** | 56.80 | 72.01 | 38.42 | **40.92** | **61.12** | 55.42 | **28.6** |

where $z_0$ is either a fixed hyperparameter or a learnable scalar clamped to a valid range. This operation preserves highly relevant scores while uniformly compressing irrelevant ones.

The shallow relevance $a_t^{(0)}$ is then used to retrieve deeper states. Under dense connections, the level-$i$ attention score is computed as

$$a_t^{(i)} = a_t'^{(i)} \cdot \hat{a}_t^{(i)} = \left( a_t^{(i-1)} R_t^{(i-1,i)} \right) \cdot \left( \sigma \left( q_t S_t^{K(i)\top} \right) W^{(i)} \right),$$

where $R_t^{(i-1,i)}$ denotes inter-level state connectivity.

However, we adopt a correlation-based dynamic routing strategy. Instead of directly multiplying $a_t'^{(i)}$ and $\hat{a}_t^{(i)}$, we suppress weak correlations via thresholded routing:

$$\hat{a}_t'^{(i)} = \hat{a}_t^{(i)} \left( \text{ReLU} \left( a_t'^{(i)\top} \hat{a}_t^{(i)} \right) - s \right),$$

where $s$ is a learnable threshold. Since $a$ is scalar-valued, an equivalent kernel-expanded formulation is used in practice:

$$\hat{a}_t'^{(i)} = w_a \, \text{ReLU} \left( \hat{a}_t^{(i)} - s \right),$$

where $w_a$ is a scalar normalization factor. Although higher-dimensional relevance metrics (e.g., $R_t^{(0,1)\top} S_t^{K(0)}$ versus $S_t^{K(1)}$) are more expressive, we adopt this formulation to balance computational efficiency and performance.

### 3.3. Recursive Sparse Attention (RSA)

We now describe the full RSA memory update and readout process. For completeness, the full formulation is presented in Appendix A.

**Sequential formulation.** First, we assume that all state tensors at time step 0 are initialized to all zeros. When initializing a zero vector as the denominator, a small scalar eps is added to prevent errors. For each level $i$, the memory length accumulator, key and value states are updated as

$$l_t^{(i)} = l_{t-1}^{(i)} + \bar{f}^{(i)} \left( k_t^{(i)} S_{t-1}^{K(i)\top} \right)^\top.$$

$$S_t^{K(i)} = S_{t-1}^{K(i)} - \frac{1}{l_t^{(i)}}^\top \bar{f}^{(i)}(\cdot) S_{t-1}^{K(i)} + \frac{1}{l_t^{(i)}} \bar{f}^{(i)}(\cdot)^\top k_t^{(i)},$$

$$S_t^{V(i)} = S_{t-1}^{V(i)} - \frac{1}{l_t^{(i)}}^\top \bar{f}^{(i)}(\cdot) S_{t-1}^{V(i)} + \frac{1}{l_t^{(i)}} \bar{f}^{(i)}(\cdot)^\top v_t^{(i)},$$

Inter-level projections and relation matrix are computed as

$$k_t^{(i+1)} = \text{MLP} \left( k_t^{(i)} - \text{SA}(k_t^{(i)}, S_t^{K(i)}, S_t^{K(i)}) \right),$$

$$v_t^{(i+1)} = \text{MLP} \left( v_t^{(i)} - \text{SA}(k_t^{(i)}, S_t^{K(i)}, S_t^{V(i)}) \right).$$

$$R_t^{(i,i+1)} = \left( 1 - \frac{1}{l_t^{(i)}} \bar{f}^{(i)}(\cdot) \right) R_{t-1}^{(i,i+1)} + \frac{1}{l_t^{(i)}} r_t^{(i,i+1)},$$

$$r_t^{(i,i+1)} = \bar{f}^{(i)}(\cdot)^\top \bar{f}^{(i+1)}(\cdot) \in \mathbb{R}^{m_i \times m_{i+1}}.$$

Sparse relevance scores are computed as

$$a_t^{(i)} = \prod_{j=0}^{i} \left( w_{a^i} \text{ReLU} \left( a_t^{(i)} - \frac{1}{z_i} \right) + \frac{1}{z_i m_i} \right) W_r^{(i)}.$$

The final output is obtained by normalized multi-level aggregation:

$$o_t = \sum_{i=0}^{c} \frac{a_t^{(i)}}{\sum_{j=0}^{c} a_t^{(j)}} S_t^{V(i)}.$$

All parameters $W^{(i)}, W_r^{(i)}, z_i,$ and $\text{MLP}^{(i)}(\cdot)$ are learnable.

## 4. Experiments

We train autoregressive language models in a token-by-token manner across multiple parameter scales. Our evaluation compares RSA against state-of-the-art linear attention models, including MetaLA, DeltaNet, Gated DeltaNet, Titans, and RWKV-7; state-space models (SSMs) such as Mamba and Mamba2; and standard Transformer baselines, including models trained from scratch under the Qwen architecture as well as models initialized from pretrained Qwen weights. Both training-from-scratch and weight-based fine-tuning experiments are conducted using randomly sampled data from the open-source FineWeb-Edu dataset. All experiments are trained on a cluster of 16 NVIDIA H100 GPUs with 80GB memory each.

### 4.1. Training Configurations

**Training from Scratch.** The RSA-360M model ($\sim$360M parameters) is trained on 15B tokens. We use AdamW with a warmup of 100 steps, processing 0.5M tokens per step, for a total of 30,000 steps. The best checkpoint is selected from steps 29,600–30,400. The peak learning rate is $4 \times 10^{-4}$ with cosine annealing, and the sequence length is fixed at 2K tokens. **RSA-360M-q.** This variant adds an additional query branch at the readout stage (A.1) attending to both key and value states, improving short-context performance.

The RSA-1.3B model ($\sim$1.3B parameters) is trained on 100B tokens. We use AdamW with weight decay 0.01 and a 200-step warmup. Each step processes 2M tokens for a total of 50,000 steps. The peak learning rate is $4 \times 10^{-4}$ (cosine annealing), with a sequence length of 2K tokens. Both models use the Mistral tokenizer (vocab size 32,000) and a four-level memory hierarchy with state size 64 per level. For comparison, Gated DeltaNet is trained from scratch with total state size 256.

**Fine-tuning from Pretrained Transformers.** We fine-tune RSA by converting pretrained Transformer attention into RSA: **Qwen-2.5-3B** is fine-tuned on 100B tokens and **Qwen-2.5-7B** is fine-tuned on 20B tokens. LoRA is applied to attention and MLP layers with rank 512 and $\alpha = 2 \times$ rank. Embedding layers and all new RSA parameters are fully unfrozen and each step processes 0.5M tokens. AdamW with a constant learning rate $1 \times 10^{-4}$ is used, and gradient clipping is applied with norm 1.0. Sequence length is set to 8K tokens, retaining the original Qwen tokenizer. RSA em-

ploys four memory levels with state size 64; for comparison, GSA uses total state size 256.

**Observations.** Across all scales, RSA matches or exceeds existing linear models and Transformer baselines on common-sense reasoning benchmarks (Table 1). Linear models such as Titans and Gated DeltaNet often converge quickly on perplexity and perform well on short-context tasks, particularly at smaller parameter scales. However, on long-horizon tasks (e.g., MMLU), these models exhibit a clear performance gap compared to RSA and Transformers. For example, the 400M RWKV-7 model achieves 26.1 accuracy on MMLU after training on 3T tokens, lower than RSA-360M trained on only 15B tokens.

These results suggest that existing linear attention models may overemphasize short-context optimization, implicitly converging toward sliding-window attention, which limits their long-range reasoning capabilities.

### 4.2. Retrieval Tasks

Table 3 reports the performance of RSA-360M and RSA-1.3B models trained from scratch on standard retrieval and recall benchmarks. Owing to the existence of multiple states and an effectively exponential memory capacity induced by state equivalence, our method consistently outperforms traditional single-state models such as GLA and Gated DeltaNet, as well as mixture-based approaches like MoM whose state utilization is relatively inefficient. These results demonstrate that RSA is substantially more effective at preserving and exploiting long-range information in retrieval-oriented tasks.

Furthermore, Table 4 presents the evaluation results of our fine-tuned models on the widely used Passkey task. We fine-tune the models with a sequence length of 8K tokens, and observe that RSA-7B-Qwen still achieve *100% retrieval accuracy* when evaluated at a significantly longer context length of 32K. This indicates that our approach is capable of attending to richer global information, rather than being overly biased toward local windowed context as is common in conventional linear models. In contrast, models such as GLA exhibit a sharp degradation in retrieval performance once the sequence length exceeds their effective state capacity (e.g., 256), and similar behavior is also observed for GSA, especially when their dynamic gating coefficients are set to small values.

### 4.3. Training and Inference Efficiency

We evaluate the training and inference efficiency of RSA under different state configurations. In our experiments, RSA adopts a four-layer design with a state size of 64 per layer. Under this setting, its overall runtime is comparable to that of GSA with a single state size of 256. Due to the introduc-

*Table 2.* Fine-tuning results of replacing Attention with RSA in Qwen-2.5 models.

| Model | Tokens | PIQA acc↑ | Hella. acc↑ | Wino. $acc_n$ ↑ | ARC-e acc↑ | ARC-c acc↑ | SCIQ $acc_n$ ↑ | MMLU acc↑ | Avg acc↑ |
|---|---|---|---|---|---|---|---|---|---|
| RWKV6-World2.1-3B* | 2.5T | 76.4 | 68.4 | 66.3 | 71.2 | 35.6 | 92.2 | 28.3 | 62.20 |
| Llama3.2-3B* | 15.0T | 76.7 | 73.6 | 69.9 | 74.5 | 42.2 | 95.7 | 56.5 | 69.87 |
| Qwen2.5-3B* | 18.0T | 78.6 | 73.5 | 68.5 | 77.4 | 45.0 | 96.2 | 65.7 | 72.13 |
| RWKV7-World3-2.9B* | 5.6T | 79.7 | 76.4 | 72.8 | 81.0 | 48.0 | 95.0 | 55.0 | 72.56 |
| GSA-3B-Qwen | 10B | 72.8 | 70.1 | 65.5 | 73.6 | 41.2 | 92.0 | 30.6 | 63.69 |
| RSA-3B-Qwen (Ours) | **10B** | 78.9 | 73.6 | 68.0 | 77.8 | 45.1 | 95.2 | 46.6 | 69.32 |
| RSA-3B-Qwen (Ours) | **100B** | **79.8** | 75.1 | 72.4 | 79.6 | 46.5 | **96.2** | 55.8 | 72.20 |
| RWKV6-7B* | 1.4T | 78.4 | 75.2 | 68.5 | 73.6 | 44.0 | 95.5 | 43.9 | 68.44 |
| Mamba-7B* | 1.2T | 81.0 | 77.8 | 72.3 | 77.6 | 46.8 | 95.9 | 33.2 | 69.23 |
| Mistral-7B* | >2T | 80.6 | 81.1 | 74.0 | 80.8 | 54.0 | 95.9 | 62.4 | 75.54 |
| SUPRA* | 100B | 79.9 | 77.1 | 70.3 | 76.0 | 45.7 | 95.0 | 34.1 | 68.30 |
| GSA-7B-Mistral* | 100B | 78.9 | 77.9 | 72.6 | 76.0 | 46.9 | 95.2 | 38.1 | 69.37 |
| Llama3.1-8B* | 15.0T | 80.1 | 78.8 | 73.6 | 81.5 | 53.7 | 96.2 | 65.0 | 75.56 |
| Qwen2.5-7B* | 18.0T | 78.8 | 79.0 | 73.1 | 80.4 | 51.4 | 96.6 | 74.2 | 76.21 |
| RSA-7B-Qwen (Ours) | **20B** | 79.6 | 80.1 | 73.8 | **82.1** | 52.6 | 96.2 | 59.1 | 74.79 |

*Table 3.* Performance of models trained from scratch on standard retrieval and recall benchmarks. All models are trained with 360M parameters on 15B tokens (top) or 1.3B parameters (bottom). Higher scores indicate better performance.

| Model | FDA | SWDE | SQuAD | NQ | TriviaQA | DROP | Avg. |
|---|---|---|---|---|---|---|---|
| **360M Parameters / 15B Tokens** | | | | | | | |
| Transformer++* | 46.14 | 25.87 | 33.22 | 18.94 | 45.97 | 20.03 | 31.70 |
| RetNet* | 5.90 | 9.28 | 22.41 | 6.91 | 40.05 | 18.59 | 17.19 |
| HGRN2* | 11.53 | 17.34 | 24.08 | 12.67 | 43.84 | 17.35 | 21.14 |
| GLA* | 11.26 | 16.78 | 27.85 | 12.77 | 43.90 | 17.68 | 21.71 |
| GSA* | 6.36 | 16.87 | 21.90 | 14.60 | 42.18 | 16.72 | 19.77 |
| Gated DeltaNet* | 20.53 | 23.24 | 28.55 | 14.98 | 44.91 | 16.48 | 24.78 |
| MoM* | 22.98 | 29.90 | 29.69 | 16.60 | 48.82 | 20.99 | 28.16 |
| RSA (Ours) | 32.64 | 28.82 | 31.44 | **18.96** | 49.68 | 20.65 | **30.37** |
| **1.3B Parameters** | | | | | | | |
| Transformer++ | 44.32 | 32.43 | 42.59 | 24.49 | 58.47 | 21.56 | 37.31 |
| RetNet* | 13.62 | 22.59 | 33.46 | 15.43 | 53.79 | 19.79 | 26.45 |
| HGRN2* | 12.35 | 23.24 | 33.19 | 19.10 | 55.27 | 19.65 | 27.13 |
| GLA+* | 27.61 | 30.93 | 35.04 | 22.27 | 56.28 | 19.45 | 31.93 |
| GSA* | 23.25 | 32.80 | 35.57 | 22.96 | 57.05 | 20.65 | 32.05 |
| Gated DeltaNet* | 30.25 | 27.65 | 34.06 | 23.22 | 58.23 | 20.36 | 32.30 |
| MoM* | 41.14 | 34.30 | 37.08 | 24.11 | 58.59 | 21.03 | 36.04 |
| RSA (Ours) | 43.53 | **35.26** | 40.09 | 24.01 | **59.42** | 20.99 | 37.21 |

*Table 4.* Passkey retrieval accuracy of fine-tuned models under varying context lengths. All models are fine-tuned with a context length of 8K tokens.

| Model / Length | 1K | 2K | 4K | 8K | 16K | 32K |
|---|---|---|---|---|---|---|
| Qwen2.5-3B | 1.0 | 1.0 | 1.0 | 1.0 | 1.0 | 1.0 |
| Qwen2.5-7B | 1.0 | 1.0 | 1.0 | 1.0 | 1.0 | 1.0 |
| GLA-7B-Qwen | 0.9 | 0.7 | 0.5 | 0.2 | 0.0 | 0.1 |
| GSA-7B-Qwen | 0.9 | 0.8 | 0.5 | 0.3 | 0.1 | 0.1 |
| RSA-3B-Qwen (Ours) | 1.0 | 1.0 | 1.0 | 1.0 | 1.0 | 1.0 |
| RSA-7B-Qwen (Ours) | 1.0 | 1.0 | 1.0 | 1.0 | 1.0 | 1.0 |

*Table 5.* Training time and peak memory consumption under different sequence configurations. All results are measured per iteration.

| Model | 32 × 2K | | 8 × 8K | | 2 × 32K | |
|---|---|---|---|---|---|---|
| | Time (s) | Memory (MiB) | Time (s) | Memory (MiB) | Time (s) | Memory (MiB) |
| RSA 64 × 4 | 5.56 | 63272 | 5.58 | 63274 | 5.56 | 63271 |
| RSA 32 × 4 | 4.21 | 60536 | 4.23 | 60536 | 4.22 | 60539 |
| GSA 256 | 4.02 | 55266 | 4.03 | 55269 | 4.06 | 55272 |
| GSA 128 | 2.98 | 53830 | 2.98 | 53831 | 3.03 | 53834 |

tion of sparse connections and sequential (recursive) reading across memory states, RSA incurs additional overhead. As a result in Table 5, the final training speed of RSA is approximately $1.4\times$ slower than GSA with an equivalent total state capacity. The same ratio is observed during inference in Fig 2, indicating consistent computational characteristics across training and deployment.

### 4.4. Long Sequence Evaluation on LongBench

We further evaluated our model on long-sequence tasks using the LongBench benchmark, as shown in Table 6.

### 4.5. Ablation Study

We conducted ablation experiments to analyze the contributions of each component in RSA. Among the variants, *Deep–Shallow Memory* exhibits the most significant im-

provement. The results are summarized in Table 7.

## 5. Conclusion

In summary, from a neuroscience-inspired perspective, we have designed an efficient memory mechanism: *deep-shallow memory states*. This mechanism is capable of representing a number of learning states on an exponential scale, substantially enhancing the representational capacity of current linear models. Complementary to this, we also propose a brain-inspired retrieval mechanism: *sparse correlation-based readout*. This mechanism is able to filter out a large amount of irrelevant noise during the retrieval process, thereby achieving a capability akin to standard attention mechanisms that focus selectively on relevant information. Together, these two contributions provide a principled approach to significantly increasing both the efficiency and expressiveness of memory-based learning systems.

## Impact Statement

This paper presents work whose goal is to advance the field of Machine Learning. There are many potential societal consequences of our work, none which we feel must be specifically highlighted here.

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

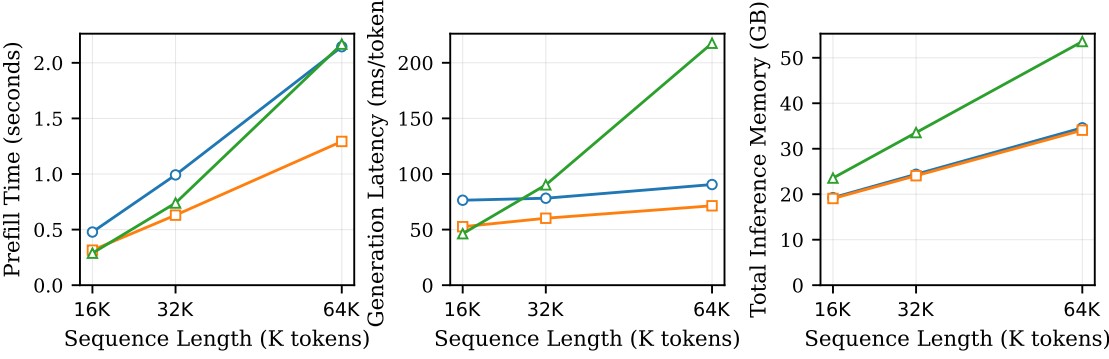

*Figure 2.* Inference performance of **Mistral-7B** under different attention mechanisms. The green triangle represents FlashAttention2, the blue circle represents RSA-16 × 4, and the orange square represents GSA-64.

## A. Parallel Approximation Formulation

*Table 6.* Performance on LongBench tasks. "Avg." denotes the average score across all tasks. Models marked with * are reported from prior works.

| Model | SQA | MQA | Sum | FS | Code | Avg. |
|---|---|---|---|---|---|---|
| RetNet-1.3B* | 9.23 | 7.23 | 6.30 | 15.76 | 40.52 | 15.81 |
| HGRN2-1.3B* | 7.38 | 6.02 | 6.51 | 15.50 | 40.11 | 15.10 |
| GSA-1.3B* | 8.21 | 6.63 | 7.75 | 20.29 | 42.83 | 17.14 |
| Gated DeltaNet-1.3B* | 8.52 | 6.61 | 7.14 | 18.00 | 41.52 | 16.36 |
| MoM-1.3B* | 8.14 | 7.11 | 6.89 | 21.26 | 47.79 | 18.24 |
| Gated DeltaNet-1.3B | 15.01 | 11.42 | 11.02 | 22.76 | 36.82 | 19.80 |
| RSA-1.3B | 16.12 | 11.39 | 10.23 | 26.72 | 43.35 | 21.56 |
| Qwen2.5-7B | 28.60 | 28.71 | 17.85 | 69.51 | 58.60 | 40.65 |
| RSA-7B-Qwen (Ours) | 25.25 | 23.68 | 16.96 | 68.96 | 58.00 | 38.57 |

*Table 7.* Ablation study of RSA on multiple benchmarks. "Avg." denotes the average score across all listed tasks.

| Model | LMB | PIQA | Hella. | Wino. | ARC-e | ARC-c | Avg. | MMLU |
|---|---|---|---|---|---|---|---|---|
| RSA-360M | 32.89 | 66.50 | 39.24 | 52.04 | 56.70 | 27.81 | 45.86 | 26.4 |
| w/o Sparse read (w/ Intensive reading = PLA) | 31.65 | 66.20 | 38.97 | 51.89 | 55.96 | 28.24 | 45.49 | 26.3 |
| w/o Sparse and Intensive reading = MVA | 30.89 | 65.90 | 38.09 | 51.64 | 56.48 | 27.39 | 45.07 | 25.6 |
| w/o Deep–Shallow Memory (= GSA) | 29.96 | 64.28 | 37.68 | 51.02 | 54.97 | 26.43 | 44.06 | 24.7 |

The parallel version of our method employs an almost lossless approximation. For ease of understanding, we represent it in operator form as follows:

$$L_t^{(i)} = L_{t-c}^{(i)} + M_L\, \bar{f}^{(i)}\left(K_{t-c:t}^{(i)} S_{t-c}^{K(i)\top}\right)^\top, \tag{2}$$

$$S^{K(i)}_{t} = S^{K(i)}_{t-c} - \left(\frac{1}{L_t^{(i)}}\right)^\top \bar{f}^{(i)}\left(K_{t-c:t}^{(i)} S_{t-c}^{K(i)\top}\right) S^{K(i)}_{t-c}$$

$$+ \frac{1}{L_t^{(i)}} \bar{f}^{(i)}\left(K_{t-c:t}^{(i)} S_{t-c}^{K(i)\top}\right)^\top K_{t-c:t}^{(i)}, \tag{3}$$

$$S^{V(i)}_{t} = S^{V(i)}_{t-c} - \left(\frac{1}{L_t^{(i)}}\right)^\top \bar{f}^{(i)}\left(K_{t-c:t}^{(i)} S_{t-c}^{K(i)\top}\right) S^{V(i)}_{t-c}$$

$$+ \frac{1}{L_t^{(i)}} \bar{f}^{(i)}\left(K_{t-c:t}^{(i)} S_{t-c}^{K(i)\top}\right)^\top V_{t-c:t}^{(i)}. \tag{4}$$

$$K^{(i+1)}_{t-c:t} = \text{MLP}\left(K^{(i)}_{t-c:t} - \text{SA}\left(K^{(i)}_{t-c:t}, S^{K^{(i)}}_{t-c}, S^{K^{(i)}}_{t-c}\right)\right), \tag{5}$$

$$V^{(i+1)}_{t-c:t} = \text{MLP}\left(V^{(i)}_{t-c:t} - \text{SA}\left(K^{(i)}_{t-c:t}, S^{K^{(i)}}_{t-c}, S^{K^{(i)}}_{t-c}\right)\right). \tag{6}$$

where $M_L$ is a lower-triangular matrix with all entries in the lower half set to 1.

The local attention outputs are computed as follows:

$$\hat{A}^{(i)}_t = \text{LA}\left(Q^{(i)}_{t-c:t}, K^{(i)}_{t-c:t}, \bar{f}^{(i)}\left(K^{(i)}_{t-c:t} S^{K^{(i)\top}}_{t-c}\right), L^{(i)}_t\right)$$

$$= Q^{(i)}_{t-c:t}\left(S^{K^{(i)}}_{t-c} - \left(\frac{1}{L^{(i)}_t}\right)^\top \bar{f}^{(i)}\left(K^{(i)}_{t-c:t} S^{K^{(i)\top}}_{t-c}\right) S^{K^{(i)}}_{t-c}\right)^\top$$

$$+ \left(Q^{(i)}_{t-c:t} K^{(i)\top}_{t-c:t} \odot M\right)\left(\frac{1}{L^{(i)}_t} \bar{f}^{(i)}\left(K^{(i)}_{t-c:t} S^{K^{(i)\top}}_{t-c}\right)^\top\right)^\top. \tag{7}$$

$$A^{(i)}_{t-c:t} = \text{LA}\left(\hat{A}^{(i)}_t, \bar{f}^{(i)}\left(K^{(i)}_{t-c:t} S^{K^{(i)\top}}_{t-c}\right), \bar{f}^{(i+1)}\left(K^{(i+1)}_{t-c:t} S^{K^{(i+1)\top}}_{t-c}\right), L^{(i)}_t\right), \tag{8}$$

$$A^{(i)}_{t-c:t} = \sigma\left(A^{(i)}_{t-c:t}\right) W^{(i)}, \tag{9}$$

$$\tilde{A}^{(i)}_{t-c:t} = \prod_{j=0}^{i}\left(w_{a^i} \text{ReLU}\left(A^{(i)}_t - \frac{1}{z_i}\right) + \frac{1}{z_i m_i}\right) W^{(i)}_r, \tag{10}$$

$$O^{(i)}_{t-c:t} = \text{LA}\left(\tilde{A}^{(i)}_{t-c:t}, \bar{f}^{(i)}\left(K^{(i)}_{t-c:t} S^{K^{(i)\top}}_{t-c}\right), V^{(i)}_{t-c:t}, L^{(i)}_t\right). \tag{11}$$

The operator LA can be replaced with GLA to incorporate data-dependent dynamic decay. The approximation primarily arises in the retrieval of $S^{K^{(i)}}_{t-c}$ using $K^{(i)}_{t-c:t}$, where we optionally include the information from the current window; however, the performance gain is minimal, and omitting this computation slightly improves efficiency.

### A.1. Incorporating Short-Sequence Reading and RSA-q

Incorporating the short-sequence read branch from MVA improves performance on short-sequence tasks, but may slightly degrade long-sequence performance. This trade-off is common among linear models: adding short-sequence enhancement mechanisms, such as gating, improves convergence by mimicking sliding-window Transformers, but limits the ability to handle tasks that require information beyond the window. The extrapolation ability of these linear models is similar to SWA, where the model focuses primarily on windowed information, making it relatively insensitive to information outside the window, thus allowing long sequences to maintain near-original perplexity.

For RSA-q, the computation requires only an additional linear projection for $q$ when reading $V$, leaving all other computations identical:

$$o_t = \sum_{i=0}^{c} \frac{a^{(i)}_t}{\sum_{i=0}^{c} a^{(i)}_t} S^{V^{(i)}}_t + \frac{q_t W^{(i)}_{qr}}{16} S^{V^{(i)}}_t, \quad W^{(i)}_{qr} \in \mathbb{R}^{d_q \times m_i}. \tag{12}$$

The additional $q$-read branch can also be applied before the sparse activation of $a^{(i)}_t$. On the 340M model, this has minimal effect on performance, hence we adopt the above training strategy.

## B. Representational Capacity

**Theorem B.1** (Representational Capacity of Compositional Key States). *Given $n$ key states, where the $i$-th state has cardinality $m_i$, the total number of distinct KV token types representable by these states is*

$$\prod_{i=1}^{n} m_i.$$

### B.1. Formal Definitions

We begin by precisely defining the concepts involved in the theorem.

**KV token type.** A KV token type refers to a unique key–value pattern in the model, which can be encoded as a discrete identifier.

**Key state.** The $i$-th key state is denoted by $S^{(i)}$, which is a finite set with cardinality

$$|S^{(i)}| = m_i.$$

Each state encodes input information at a specific level of abstraction or along a particular feature dimension.

**Representability.** A KV token type is said to be representable if it can be uniquely identified by an $n$-tuple

$$(s^{(1)}, s^{(2)}, \ldots, s^{(n)}), \quad \text{where } s^{(i)} \in S^{(i)}.$$

The overall representational space of the system is therefore the Cartesian product

$$\mathcal{R} = S^{(1)} \times S^{(2)} \times \cdots \times S^{(n)} = \left\{ (s^{(1)}, \ldots, s^{(n)}) \mid s^{(i)} \in S^{(i)} \; \forall i \right\}.$$

### B.2. Achievability via Combinatorial Counting

We first show that $\prod_{i=1}^{n} m_i$ distinct KV token types are *achievable*. The proof proceeds by mathematical induction on the number of states $n$.

**Base case ($n = 1$).** When there is only a single state $S^{(1)}$, the representable KV token types are in one-to-one correspondence with the elements of $S^{(1)}$. Since $|S^{(1)}| = m_1$, the number of representable types is exactly

$$m_1 = \prod_{i=1}^{1} m_i.$$

Thus, the base case holds.

**Inductive hypothesis.** Assume that for a system with $n = k$ states, the total number of representable KV token types is

$$\prod_{i=1}^{k} m_i.$$

**Inductive step ($n = k + 1$).** Consider a system with $k + 1$ states. Its representational space can be written as

$$\mathcal{R} = \left( S^{(1)} \times \cdots \times S^{(k)} \right) \times S^{(k+1)}.$$

Let

$$\mathcal{R}_k = S^{(1)} \times \cdots \times S^{(k)}$$

denote the intermediate representational space formed by the first $k$ states. By the inductive hypothesis,

$$|\mathcal{R}_k| = \prod_{i=1}^{k} m_i.$$

For each intermediate representation in $\mathcal{R}_k$, combining it with the $(k + 1)$-th state $S^{(k+1)}$ yields $m_{k+1}$ distinct final representations. Since this combination is independent for each element of $\mathcal{R}_k$, the total number of representable KV token types is

$$|\mathcal{R}| = |\mathcal{R}_k| \cdot |S^{(k+1)}| = \left( \prod_{i=1}^{k} m_i \right) m_{k+1} = \prod_{i=1}^{k+1} m_i.$$

By mathematical induction, the theorem holds for all positive integers $n$.

### B.3. Information-Theoretic Converse (Upper Bound)

We now show that $\prod_{i=1}^{n} m_i$ is also an upper bound, and hence the result is tight.

Each key state $S^{(i)}$ can encode at most $\log_2 m_i$ bits of information. Assuming the states are jointly compositional and distinguishable, the total information capacity of the system is

$$C_{\text{total}} = \sum_{i=1}^{n} \log_2 m_i = \log_2 \left( \prod_{i=1}^{n} m_i \right).$$

By basic information-theoretic principles, a system with capacity $C_{\text{total}}$ bits cannot represent more than $2^{C_{\text{total}}}$ distinct symbols. Therefore, the total number of distinguishable KV token types satisfies

$$|\mathcal{R}| \leq 2^{C_{\text{total}}} = \prod_{i=1}^{n} m_i.$$

### B.4. Conclusion

Combining the achievability result from combinatorial counting with the information-theoretic upper bound, we conclude that the maximum number of distinct KV token types representable by $n$ key states of sizes $\{m_i\}_{i=1}^{n}$ is exactly

$$\prod_{i=1}^{n} m_i.$$

## C. Formal Proof of Theorem 2

**Definition C.1** (KV Token Type Representation System)**.** An $n$-layer state-based representation system consists of:

- An input space $\mathcal{X} \subseteq \mathbb{R}^d$, representing all possible input vectors (e.g., embedded tokens).

- For each layer $i \in \{1, \ldots, n\}$, a discrete state space $\mathcal{S}_i$ with cardinality $|\mathcal{S}_i| = m_i$, which stores information accumulated from the input history.

- A representation function
$$\Phi : \mathcal{X} \times \mathcal{S}_1 \times \cdots \times \mathcal{S}_n \to \{1, \ldots, N\},$$
mapping the current input vector and all layer states to a discrete token type identifier.

At time step $t$, upon receiving input $x_t$, the system performs:

1. Token type computation:
$$id_t = \Phi(x_t, S_{t-1}^{(1)}, \ldots, S_{t-1}^{(n)}).$$

2. State update for each layer:
$$S_t^{(i)} = \text{Update}^{(i)}(x_t, id_t, S_{t-1}^{(i)}).$$

**Definition C.2** (Maximum Capacity)**.** The *maximum capacity* $C_{\max}$ of a representation system is defined as the supremum of the number of mutually distinct token type identifiers that the system can generate over all sufficiently rich infinite input sequences.

By Theorem 1 (the combinatorial counting principle), the theoretical maximum capacity is

$$C_{\max} = \prod_{i=1}^{n} m_i.$$

Our goal is to analyze the computational conditions under which this bound is achievable.

**C.1. Capacity Achievability of the Recursive Structure**

We show that the recursive computation proposed in Theorem 2, when coupled with an appropriate state update mechanism, achieves the full capacity $\prod_{i=1}^{n} m_i$.

- **State Update.** For each layer $i$, the key state $S_t^{K(i)} \in \mathbb{R}^{m_i \times d_k}$ consists of $m_i$ prototype vectors (cluster centers). The update follows a competitive online clustering rule (e.g., attention-based Winner-Take-All or a delta-rule update), ensuring that each input $k_t^{(i)}$ is assigned to exactly one cluster.

- **Representation Computation.** The input $x_t$ is first linearly projected to obtain $k_t^{(1)}$. For $i = 1, \ldots, n-1$, define recursively:
$$k_t^{(i+1)} = \mathrm{MLP}^{(i)}\left( k_t^{(i)} - \mathrm{SA}\left( k_t^{(i)}, S_t^{K(i)}, S_t^{K(i)} \right) \right),$$
where
$$\mathrm{SA}(q, K, V) = \mathrm{Softmax}\left( \frac{qK^\top}{\sqrt{d}} \right) V.$$

- **Token Type Identification.** The token type identifier is given by the joint cluster assignments:
$$id_t = (c_1, \ldots, c_n), \quad c_i = \arg\max \mathrm{Softmax}\left( \frac{k_t^{(i)} S_t^{K(i)\top}}{\sqrt{d}} \right).$$

**Step 1: First-layer Partitioning.** The first-layer state $S^{K(1)}$ partitions the projected input space into $m_1$ regions, assigning each $k_t^{(1)}$ to a cluster index $c_1 \in \{1, \ldots, m_1\}$. This realizes a coarse, first-level classification of the input.

**Step 2: Residual Extraction and Approximate Orthogonality.** The attention term
$$\mathrm{SA}(k_t^{(1)}, S^{K(1)}, S^{K(1)})$$
constitutes the optimal reconstruction of $k_t^{(1)}$ in the subspace spanned by the prototype vectors, under the softmax-weighted inner-product metric. Consequently, the residual
$$r_t^{(1)} = k_t^{(1)} - \mathrm{SA}(\cdot)$$
captures information not represented by the first-layer clustering, corresponding to components approximately orthogonal to the subspace encoding $c_1$.

**Step 3: Induced Hierarchical Partitioning.** The residual $r_t^{(1)}$ is transformed by $\mathrm{MLP}^{(1)}$ to produce $k_t^{(2)}$. Since $r_t^{(1)}$ is approximately decorrelated from the first-layer cluster assignment $c_1$, the induced representation $k_t^{(2)}$ carries information that is approximately independent of $c_1$. The second-layer state $S^{K(2)}$ therefore performs a clustering that is conditionally independent of the first-layer decision. This process recurses through all layers.

**Step 4: Capacity Counting.** Each layer $i$ independently produces one of $m_i$ cluster assignments based on information extracted from orthogonal residual components. Thus, the joint assignment
$$(c_1, \ldots, c_n)$$
ranges over all combinations in $\mathcal{S}_1 \times \cdots \times \mathcal{S}_n$. The total number of distinct token types is therefore
$$\prod_{i=1}^{n} m_i.$$

For example, the above process can be viewed as a recursive division of large grids into smaller ones, where each vector can then be approximated.

**C.2. A Constructive Approximation Result**

We first formalize the system model and then present a non-rigorous but constructive argument showing that any computation process achieving the capacity upper bound in Theorem 1 can be functionally approximated by the recursive formulation described in Theorem 2.

C.2.1. PROBLEM FORMULATION AND DEFINITIONS

**Notation and Setup.** Let $\mathcal{K}$ denote the input space of all possible key vectors $k$. An *n-level key-state system* consists of $n$ state sequences $\{S_t^{(1)}, \ldots, S_t^{(n)}\}$ together with a corresponding family of encoding functions $\{f^{(i)}\}_{i=1}^n$. Each state $S_t^{(i)} \in \mathcal{S}_i$ takes values in a finite set of cardinality $m_i$, i.e., $|\mathcal{S}_i| = m_i$.

The *joint state* of the system at time $t$ is defined as

$$\mathbf{S}_t = (S_t^{(1)}, \ldots, S_t^{(n)}) \in \mathcal{S} := \mathcal{S}_1 \times \cdots \times \mathcal{S}_n.$$

The system induces an *instantaneous encoding function*

$$\Phi_t : \mathcal{K} \to \mathcal{S}, \qquad \Phi_t(k_t) = \mathbf{S}_t,$$

which is determined by the system history up to time $t - 1$ and the current input $k_t$.

The number of *distinct KV token types* representable by the system is defined as the cardinality of the image of $\Phi_t$, denoted by $|\mathrm{Im}(\Phi_t)|$. By Theorem 1, this quantity is upper bounded by $\prod_{i=1}^n m_i$.

**Statement of Theorem 2.** Theorem 2 asserts that if the instantaneous encoding function $\Phi_t$ achieves the maximal capacity, namely

$$|\mathrm{Im}(\Phi_t)| = \prod_{i=1}^n m_i \quad \text{(for sufficiently rich input sequences),}$$

then there exists a functionally equivalent system whose hierarchical encoding process can be *approximated* by the following recursion:

$$k_t^{(i+1)} = \mathrm{MLP}\left(k_t^{(i)} - \mathrm{SA}\left(k_t^{(i)}, S_t^{(i)}, S_t^{(i)}\right)\right),$$

where $k_t^{(1)} = k_t$ and $S_t^{(i)}$ denotes the state at level $i$.

Here, *approximation* means that for any $\epsilon > 0$, there exist parameterizations of the MLP and SA operators such that the discrepancy between the induced joint-state distributions (e.g., under total variation distance) is less than $\epsilon$.

C.2.2. PROOF OVERVIEW

The argument proceeds in three conceptual steps:

1. **Implications of the maximal capacity condition:** achieving maximal capacity forces the system to implement a lossless, hierarchical, and layer-independent encoding scheme.

2. **Derivation of the required mathematical structure:** this condition implies that each layer must remove information already captured by previous layers and operate on the remaining residual.

3. **Universality of attention and MLPs:** standard self-attention realizes adaptive linear projections, while MLPs are universal approximators of nonlinear functions, together forming a universal approximation mechanism for such encoders.

C.2.3. STEP I: CONSEQUENCES OF MAXIMAL CAPACITY

If $|\mathrm{Im}(\Phi_t)| = \prod_{i=1}^n m_i$, then the following properties must hold.

**Surjectivity.** The mapping $\Phi_t$ must be surjective, i.e., every joint state $\mathbf{s} \in \mathcal{S}$ is attained by at least one input $k_t$.

**Layer-wise Independence.** The state variables $S_t^{(1)}, \ldots, S_t^{(n)}$ must be (approximately) statistically independent under the induced input distribution. Otherwise,

$$H(S_t^{(1)}, \ldots, S_t^{(n)}) < \sum_{i=1}^{n} H(S_t^{(i)}) \leq \sum_{i=1}^{n} \log m_i,$$

implying that the number of distinguishable joint states is strictly smaller than $\prod_i m_i$, contradicting maximal capacity.

**Lossless Hierarchical Encoding.** The encoding process can therefore be decomposed into deterministic functions:

$$S_t^{(1)} = g_t^{(1)}(k_t),$$
$$S_t^{(2)} = g_t^{(2)}(k_t, S_t^{(1)}),$$
$$\vdots$$
$$S_t^{(n)} = g_t^{(n)}(k_t, S_t^{(1)}, \ldots, S_t^{(n-1)}).$$

To preserve independence, each $g_t^{(i)}$ must depend only on information in $k_t$ that is not already captured by the preceding states, i.e., information orthogonal to previous layers.

C.2.4. STEP II: EMERGENCE OF A RESIDUAL LEARNING STRUCTURE

Let $\mathcal{H}^{(i)}$ denote the feature subspace spanned by all possible representations of the state $S_t^{(i)}$. The independence condition implies

$$\mathcal{H}^{(i)} \perp \mathcal{H}^{(j)} \qquad \text{for } i \neq j$$

under the input distribution.

Consider the transformation from $k_t^{(i)}$ to $k_t^{(i+1)}$. To ensure that $S_t^{(i+1)}$ encodes information independent of $S_t^{(i)}$, the transformation must satisfy:

**Removal of explained information.** The component of $k_t^{(i)}$ lying in $\mathcal{H}^{(i)}$ must be removed:

$$r_t^{(i)} = k_t^{(i)} - \text{Proj}_{\mathcal{H}^{(i)}}(k_t^{(i)}),$$

yielding a residual orthogonal to $\mathcal{H}^{(i)}$.

**Nonlinear refinement.** A nonlinear transformation $\psi^{(i)}$ is then applied to extract features for the next state:

$$k_t^{(i+1)} = \psi^{(i)}\left(k_t^{(i)} - \text{Proj}_{\mathcal{H}^{(i)}}(k_t^{(i)})\right).$$

Hence, any system achieving maximal capacity must follow a *projection–residual–nonlinearity* paradigm.

C.2.5. STEP III: UNIVERSAL APPROXIMATION VIA SA AND MLP

We now show that the recursion

$$k_t^{(i+1)} = \text{MLP}\left(k_t^{(i)} - \text{SA}(k_t^{(i)}, S_t^{(i)}, S_t^{(i)})\right)$$

constitutes a parametric realization of the above structure.

**Self-Attention as Adaptive Projection.** The standard attention operator is given by

$$\text{SA}(q, K, V) = \text{Softmax}\left(\frac{qK^\top}{\sqrt{d}}\right)V.$$

When $K = V = S_t^{(i)}$, the output is a convex combination of state vectors, with weights determined by similarity to $q$.

**Key observation.** For a finite set of $m_i$ prototype vectors, the operation $\text{SA}(q, S_t^{(i)}, S_t^{(i)})$ can approximate the $\ell_2$ projection of $q$ onto the convex hull spanned by $S_t^{(i)}$. Thus, it serves as a differentiable, data-adaptive approximation to $\text{Proj}_{\mathcal{H}^{(i)}}(q)$.

**MLP as a Universal Nonlinear Approximator.** By the universal approximation theorem, a sufficiently large MLP can approximate any continuous function on a compact domain. Therefore, the MLP can implement the nonlinear refinement $\psi^{(i)}$ required to extract new, independent features from the residual.

**Completeness of the Recursion.** Combining both components:

- the subtraction term computes a residual orthogonal to the current state subspace, enforcing information independence;

- the MLP transforms this residual into a representation suitable for constructing the next-level state.

Consequently, the proposed recursion precisely instantiates the mathematical structure required to achieve maximal capacity. Since self-attention and MLPs can approximate the required projection and nonlinear operators, respectively, the recursion can approximate any system realizing the same class of hierarchical encodings.

