# OpenReview forum: "RSA: Recursive Sparse Attention with Hierarchical Deep–Shallow Memory and Sparse Activation"
_ICML.cc/2026/Conference — Submitted to ICML 2026_

### Official Review · Reviewer_aRBa · 2026-03-09

**Soundness:** 2
**Presentation:** 1
**Significance:** 2
**Originality:** 2
**Overall Recommendation:** 2
**Confidence:** 3

**Summary:**

The authors propose a new, biologically motivated linear attention variant to address some of the remaining gaps between full attention and the linear counterparts. They propose to recursively store residual memories - memory layers are stacked, and the lower ones store coarser representations, while the higher ones store corrections. The memory is multi-slot, and the authors also propose sparse readout, sparsified both based on the activation strength within the bank, but also based on the correlation between neighboring states.

**Compliance With Llm Reviewing Policy:**

Affirmed.

**Final Justification:**

The paper is very hard to read. It would be basically impossible to reproduce it based on the paper alone. After pointing out undefined and unclear points in the paper, the authors, instead of addressing the complaints, chose baseless attacks against the reviewer that continued throughout the whole reviewing process. I was unable to spot any intent of constructive discussion (As an example, see the author's explanation on the key state in the last message: "Key state simply means key state"). The paper needs a significant rewrite to be presentable at any conference. Thus I am maintaining my original score.

**Key Questions For Authors:**

- Fig 1:
  - Where is k from the formulas in the picture?
  - (a), (b), (c) are not defined in the picture, yet referred to in the caption.
- A minimal description of why the MLP is needed would be nice.
- Which layers are replaced by RSA? How deep is the recursion in the RSA? Does each of the recursive passes also add an MLP? How big is this MLP compared to the normal Transformer layers?
- How many extra parameters do the RSA layers add to the model? Do you compensate for the extra parameters by removing layers when replacing the original attention with RSA?
- What do the terms I mentioned in the weaknesses mean?
- In L105, the citation Kveim et al. is used as a general citation for neuroscience without specifying how it is related.
- In L107, right, the attention formula is invalid: the M mask is additive (with 0 and -inf values), not multiplicative.
- In L137-L144, the sliding window attention cannot be described as a simple marginalized selection on the indices, indicated by the formulas in L137, because each row of the attention matrix requires a different subset of selected indices. Additionally, i is not defined in L144.
- In L163, the axis of the top-k is not specified.
- In L110, s is not defined.
- In L201, what is "multi-scale compressed noise"? How can the noise be compressed? Why? What part of the method is the "multi-scale compressed noise?
- L224, right: What does it mean that "weakly correlated information is compressed or filtered"? What is the "later regression"?
- Replacing parts of a formula with a \cdot is very confusing. Please consider defining the missing part as a variable and substituting it into the formula when needed.
- The -q trick in L361 seems to be critical for good performance on some of the models, yet it is not explained in the main paper. A minimal intuition on why this is done would be needed.

**Limitations:**

yes

**Strengths And Weaknesses:**

Strengths:
- Interesting ideas: multi-slot deltanet-style memory with multiple levels of coarseness
- Evaluated on many downstream tasks

Weaknesses:
- The paper is extremely hard to read, to the extent that it is impossible to understand what the authors did and why. The authors use fancy terms like "shallow-deep memory" (I guess it refers to the recursive nature?), "attention-concentrated readout" (I guess the sparsity part?), "discriminatively storing all information" (??), "key state" (??), etc., without explaining what they mean. They confidently claim long paragraphs on how the brain works, with ideas that some of which I have never heard before, without any citations (e.g., Sec 3.1, Intuitive Explanation paragraph). The text and Fig 1 don't match, e.g., there is no k in fig 1, but there is in the equation. at L240. Fig 1 does not have an MLP, but eqs in L315 do. The text doesn't even mention that the storage is a key-value memory until L303 (right), suggesting that it is delta rule-like, but maybe it is not? The easiest way to (partially) understand the paper is to look at Fig 1 and ignore the text.
- Evaluations: The "Transformer++" baseline is missing for the 1.3B model in Tab 1. In Tab 2, the fair evaluation should also fine-tune the baseline models on the same data, and compare them to those, but that was not done here. This confounds the effect of finetuning on new data with the effect of the architectural change. In Tab 3. "Transformer++" is better in "Avg.", yet RSA is bold. Tab 2,3, many columns are not bolded.

---

> ### Author Rebuttal · Authors · 2026-03-30
>
> # W1
> (1) shallow...: The phrase used is "deep-shallow memory." A brief explanation appears, lines 80–83; a detailed intuitive explanation in lines 209–219, lines 165–175; and the iterative formulation in line 239 onward.
>
> (2) attention-co...: This refers to focusing on relevant memory. Line 88 mentions attention dispersion in linear attention, a well-known issue. The term is followed by an accessible explanation in line 179, and Section 3.2 details how correlation-aware sparsity achieves concentration, with the retrieval process described from line 257.
>
> (3) discriminati...: This means storing information units such that they remain distinguishable, avoiding information loss. A brief explanation is provided around line 187.
>
> (4) **key state**: We believe this would not be a question if you had briefly reviewed Section 2.3. The concept of state is fundamental and indispensable in language-oriented linear attention models. This question is comparable to asking what Adam is. Specifically, we define what a state is in line 130, and in line 151 we introduce the key state in GSA as the compressed representation of keys. And other "state" information : line 187, line 192 , 151.
>
> (5) Brain...: We describe a cognitive process characteristic of certain intelligent systems, reflecting the authors' own reasoning. Such descriptions do not strictly require citations and, based on other reviewers' feedback, are generally acceptable.
>
> (6) Figure...: Figures are intended to highlight core operations, omitting details that may obscure understanding. This is common practice; for example, ICML 2025 best paper COLLABLLM and NSA similarly abstract details in figures while providing full equations in text. Figure 1 uses a generic variable x to represent a general storage process, analogous to a function. This approach is widely used, including in the original Transformer paper. Expecting a single figure to convey all details is unrealistic; many top papers would not satisfy such a requirement. Comment that "The easiest way to (partially) understand the paper is to look at Fig 1 and ignore the text" suggests a focus on the figure at the expense of reading the text, leading to misunderstanding.
>
> (7) Key-...: Line 148 explicitly mentions compressing K and V into a state. This suggests that foundational sections were skipped. We recommend reviewing the works cited in Section 2.3.
>
> # W2
> We have added Transformer++ from GDN to Table 1.We have fine-tuned baseline models on the same data. (detail in Reviewer 39Kc)
>
> # Q
>
> 1.Refer to (6) . Regarding (a), (b), (c), thank you for pointing this out.
>
> 2. The MLP serves multiple purposes: (1) improving performance; (2) aligning with the Transformer architecture to facilitate interpretability when comparing our method to Transformers, as noted in line 208; and (3) mathematically, projecting error information into a higher-dimensional manifold and back, which helps balance feature focus and enables richer pattern storage.
>
> 3.Our architecture is purely linear; RSA replaces the attention in all layers (e.g., line 375). RSA uses a recursion depth of 4 (lines 371 and 330). Each recursive pass includes an MLP. MLP: totals 0.25M parameters, less than 0.1% of a qwen layer.
>
> 4.For from-scratch training, we compensate by using fewer heads or layers.
>
> 5.Refer to W1.
>
> 6.Our method draws mainly from our own observations and general neuroscience knowledge. Thus, it is cited only as a representative example.
>
> 7.The form is used in many top-tier papers (e.g., GLA Section 2.1, GSA Equation 1). M=Inf*(QK^\top).
>
> 8.In context, i is the index of the current Q and K tokens, and attention is applied within a window of size s around i.
>
> 9.We perform top-k along the sequence dimension of K: axis=-1
>
> 10.s is defined earlier in line 110 as the size of segment-level representations.
>
> 11.The method compresses irrelevant information at multiple scales. As described in lines 113–195: remaining irrelevant portions are compressed for retention. Section 3.2 (from line 298) explains that the most relevant attention scores are retained.
>
> 12. Weakly correlated information, reflected in the attention scores, is set to zero and replaced with a small average value 1/(zm). Irrelevant information is compressed by it.
>
> 13.We use \(\cdot\) to denote element-wise multiplication of vector.
>
> 14.We explain this in Appendix A.1.
>
> **Details: https://anonymous.4open.science/r/icml26-rsa-29B9/, Length exceeds 20,000.
> Your comments suggest you are not a specialist in our area and have engaged in noticeable skimming (e.g., missing basic knowledge of LA sec2.3). We recommend reading foundational works such as GLA, GSA, MVA.
> Finally, we respectfully ask that you reconsider our paper with a more thorough reading. Our work has undergone extensive iteration, and we believe it should not be hastily judged based on a superficial reading. While some of your points have merit, many reflect bias and a lack of substantive engagement with our approach.**

---

> > ### Author Rebuttal · Reviewer_aRBa · 2026-04-02
> >
> > I acknowledge the fact that I am not up to date with all the new works on linear attention variants, but I have multiple published papers on the topic, so I also think I am not completely uninformed. I will let the AC decide. However, I reject the accusation of skimming, as well as I reject the accusation of not reading the paper thoroughly. I spent an unreasonable amount of time reviewing this paper (reading it 3 times). I also reject the accuastion of "many reflect bias and a lack of substantive engagement with our approach." Moreover, the author's rebuttal glosses over my main concerns and intentionally tries to downplay, justifying it with what was already in the paper and was unclear in the first place. For example:
> >
> > - Line 80-83: shallow memory stores coarse information for fast access to simple queries, while deep memory stores residual fine-grained information related to the shallow content - based on my understanding, your model does a constant number of accesses, and also it is unclear to me what is a "shallow content". Lines 209-219 are a story about human memory without any citations.
> > - Line 88 mentions attention dispersion in linear attention, a well-known issue. - I am aware of this issue, hower this does not answer the term you have not defined.
> > - key state: the question was not about state. The question was about the **key** state. How is that different from the normal state.
> > - In Line 148, the authors indeed mention keys and values, but in context of a different line of work, not their own: "Another line of work compresses keys and values online while retaining softmax computation at the output stage". How one is supposed to understand that they also use or do not use one based on this?
> > - Regarding figures: yes, papers often use figures with a simplified version of their algorithm. The main difference is that many of those papers can be understood even without the figure, while this one cannot be understood even with the figure.
> > - "Brain...: We describe a cognitive process characteristic of certain intelligent systems, reflecting the authors' own reasoning. Such descriptions do not strictly require citations and, based on other reviewers' feedback, are generally acceptable." If the authors write a long paragraph about how the brain works, it either needs a citation, or it needs to state it clearly that it's the authors own idea, and not an established fact.
> >
> > Given that the authors' main strategy is to attack reviewers rather than improve their paper, I maintain my score. Specifically, I think the writing quality of this paper is below the ICML standards.

---

> > > ### Author Response · Authors · 2026-04-03
> > >
> > > Thank you for your further comments. To clarify, we have never suggested that the "... strategy is to attack reviewers." If these are seen as attacks, then your earlier phrase "ignore the text" was itself an attack and insult. **So, our and your responses  both only state facts, and we began with full gratitude for your feedback in link.** We are unsure how this conclusion was reached. Please point out any specific content that made you uncomfortable, and we will revise accordingly. After we make the revisions, we would like to ask for your further reassessment. If justified, we will offer an apology for the "specific content" you pointed out.
> > >
> > > Regarding the claim that we are trying to "rather than improve ..."—this appears to be an imposed conclusion. We have provided detailed responses to every comment, demonstrating our strong willingness to improve the paper. Given the length constraints, we included core explanations in the rebuttal and a much more detailed response (4 pages) in an anonymous link. This substantial effort reflects our genuine commitment to improvement.
> > >
> > > Furthermore, the statement "I spent an ... time reviewing ... (reading it 3 times)" together with "I am not up to date with all the new works ..." reflects an unfair reviewing practice. As an experienced reviewer of dozens of papers, I know that when lacking sufficient background in a domain, one should read the paper, then study the foundational works cited, and only then re-review multiple times. For this ICML cycle, for instance, I was assigned 6 papers with Policy A and also helped review several for my group's professor, spending over a week carefully consulting materials and providing fair feedback, including detailed weaknesses and constructive suggestions. I fully recognize that a paper represents the authors' hard work, and an unfair review could delay graduation, alter life plans, or have an indelible negative impact—just as for me. Moreover, the field evolves rapidly; failing to stay current leads to significant gaps and misjudgments.
> > >
> > > Responses to the examples you raised:
> > >
> > > 1. Lines 80-83: The current design does indeed involve a fixed number of access steps, but this does not prevent shallow memory from being accessed first and producing an output, after which that fast access aids recursive access to deeper layers. "Shallow content" refers to compressing the sequence into a fixed-size state via the formulas in line 151 or 240. Deeper layers then process the residual between the sequence and the compressed information, thereby enabling a coarse-to-fine process. Lines 209-219 provide an intuitive explanation based on our own cognitive observations, which does not require a citation. If you could suggest one, and we would be happy to follow your advice.
> > >
> > > 2. Regarding the phrase in line 178: we are willing to define it. We will add "attention-concentrated readout:" at the beginning of the first sentence of the paragraph in line 195.
> > >
> > > 3. Key state simply means key state, where "key" refers to the key in query-key-value. Following your suggestion, we will add an explicit definition in the paper.
> > >
> > > 4. This is precisely why reading the specific details of our method is necessary. We provide explicit formulations in lines 209, 240, 308, and others, along with textual explanations in line 246 and explicit mentions like "key and value states" in line 302, etc. Following your suggestion, we will add an explicit statement at the beginning indicating we use key and value states.
> > >
> > > 5. Our method is quite complex, and the figure reflects that complexity. The figure and the text are meant to complement each other: the main content is explained in the text, and the figure serves as a memory aid after reading. Based on your feedback, we are simplifying the figure to focus on the core ideas. If you have specific suggestions for improvement, we would greatly appreciate them.
> > >
> > > 6. We clearly indicated that the relevant text provides intuitive explanations and observations. We will now add a note that these reflect our own views and draw on public knowledge from neuroscience.
> > >
> > > **Finally, your assertion that "the author's rebuttal ... tries to downplay" does not hold, as we have provided detailed responses to all concerns and are making targeted revisions. Moreover, your further response suggests you still have not carefully read our paper or rebuttal. We therefore believe your conclusion that "... is below the ICML standards" is quite incorrect.**
> > >
> > > We look forward to further discussion. **Your comments is very helpful to improve our paper.** Our method is highly meaningful—**providing a linear model fully comparable to Qwen and laying a solid foundation for a next-generation architecture and training paradigm that could truly surpass standard Transformers (see Reviewer 39Kc).** Our original goal is a best nomination, but borderline now is the outcome.
> > >
> > > Finally, regardless of acceptance, we will make all discussion public for community feedback.

---

### Official Review · Reviewer_FfEX · 2026-03-11

**Soundness:** 2
**Presentation:** 3
**Significance:** 2
**Originality:** 3
**Overall Recommendation:** 3
**Confidence:** 4

**Summary:**

This paper proposes Recursive Sparse Attention (RSA), which tries to get linear attention models closer to full softmax Transformers without giving up their efficiency. The core idea has two parts. First, instead of compressing everything into a single fixed-size state, RSA uses a hierarchy of "deep" and "shallow" memory states. Second, at readout time, RSA applies a ReLU-based sparse gating to selectively attend only to the most relevant states rather than mixing everything together, which the authors argue helps filter out noise accumulated in the recurrent states. On the experimental side, the authors train 360M and 1.3B models from scratch on 15B and 100B tokens respectively, and also fine-tune Qwen-2.5 at the 3B and 7B scale. They compare against Gated DeltaNet, RWKV-7, Titans, SSM models and standard Transformers on a mix of commonsense reasoning, few-shot complex tasks, retrieval, and long sequence evaluation tasks, reporting that RSA is generally competitive with or slightly better than the strongest linear baselines and narrows the gap to full attention on most benchmarks.

**Compliance With Llm Reviewing Policy:**

Affirmed.

**Key Questions For Authors:**

Q1. Given that efficiency is the core motivation for linear attention, could you provide a more detailed breakdown of where this overhead comes from?
Q2. Can you compare against recent sparse attention approaches like with MoBA and NSA?

**Limitations:**

The paper briefly acknowledge computational overhead but downplay it. They do not discuss the fragility of cross-paper comparisons, and the absence of recent sparse attention baselines. A short paragraph explicitly listing these gaps would strengthen the paper.

**Strengths And Weaknesses:**

Strengths:
1. Well-motivated architecture with a clear intuition, the deep-shallow memory idea makes intuitive sense and the paper explains it clearly..
2. The experiments are fairly thorough and strong retrieval and passkey results. The evaluation spans across multiple type of tasks and also considered both pretraining and finetuning scenarios. The passkey retrieval experiment (Table 4) is convincing, where RSA-7B-Qwen achieves 100% accuracy at 32K context while fine-tuned on only 8K, whereas GLA and GSA degrade sharply.

Weaknesses:
1. The efficiency cost is real and glossed over. RSA is about 1.4x slower than GSA at equivalent total state capacity, which is not a "comparable" performance.
2. Some of the experiment comparisons are not so convincing. The cross-paper baselines (marked with stars in paper) use different tokenizers and data, the gaps between self-reproduced and reported numbers are sometimes large (see Gated DeltaNet on LongBench in table 6), which makes it hard to trust the rankings.
3. There is no comparison with sparse attention methods like MoBA and NSA.

---

> ### Author Rebuttal · Authors · 2026-03-30
>
> # W1
> We believe efficiency comparisons are most meaningful when performance is comparable. To this end, we trained RSA with 4 levels and state sizes of 32, and compared against GSA with a state size of 256. Our method achieves slightly better performance with substantially lower computation, demonstrating higher efficiency. (details in Q1 or more in Reviewer CQjr).
>
> Furthermore, our method is also an efficient way to scale the state capacity. The effective state capacity grows exponentially with the number of levels; specifically, our capacity is 64^4, whereas GSA has a capacity of 256. Therefore, the actual capacities are not equal. **In line 425 of the paper, we have changed "capacity" to "size" to clarify this, which was our oversight. In practice, a fair efficiency comparison with RSA would require using GSA with a capacity of 64^4.**
>
> Finally, we would like to clarify that our intention in reporting efficiency under the same state size is to help researchers in the field understand the additional overhead introduced by our modifications. And, from a capability perspective, scaling the state size of GSA, GLA, or DeltaNet still leaves certain tasks—such as very long passkey retrieval or complex MMLU tasks—infeasible. In contrast, our model handles these tasks effectively, which we view as a major advantage.
>
> # W2
> For all models trained from scratch, we used identical tokenizers and sampled data from the FineWeb-Edu dataset. We acknowledge that due to time and resource constraints, we cannot perfectly replicate the training setups of prior works. **Therefore, we fine-tuned the current best-performing linear model, Gated DeltaNet, on FineWeb-Edu and compared our model against it. Our baseline model may underperform relative to the one reported in GDN or other papers due to data limitations, but even under these circumstances, we were able to optimize RSA’s performance to exceed that of the baseline. We recommend that readers rely on all our reproduced results, including those in Table 6. The originally reported numbers serve only to indicate the existence of those methods and their relative performance; the gap between our reproduced results and them is left for readers to interpret.**
>
> Due to limited time and resources, we were unable to retrain all 1.3B-parameter baselines from scratch. Since our method shows substantial improvements over GSA, it naturally achieves stronger performance in table 6. As for why Gated DeltaNet underperforms relative to MoM in this table, we suspect it may be more sensitive to data variations.
>
> # W3
> First, our method is a linear model, and we believe it is not entirely fair to directly compare it with sparse attention methods such as MoBA and NSA, whose complexity is at least n^(4/3). However, anticipating that this argument may not be fully convincing, we conducted small-scale training from scratch for MoBA and NSA. Due to time and resource constraints, we used a parameter size of 360M. For NSA, we used the implementation from the FLA operator library, which requires the number of query heads to be a multiple of 16 times the number of key-value heads.
>
> |Model|LMB |PIQA |Hella. |Wino.|ARC-e |ARC-c |SIQA|BoolQ |Avg |
> |-|-|-|-|-|-|-|-|-|-|
> |NSA-370M|32.05|65.71|38.15|51.02|56.36|27.22|37.32|60.12|45.99|
> |MoBA-360M|32.01|66.51|39.15|51.97|56.21|26.89|36.58|60.09|46.18|
> |RSA-360M|32.89|66.50|39.24|52.04|56.70|27.81|37.24|60.12|46.57|
>
> We believe that NSA’s advantages are less pronounced at smaller scales and with limited training budgets.
>
> # Q1
> As detailed in Section 3.3 (or lines 590–630 of the appendix), our method requires three LA operator calls. Among these, the LA operation used to compute the inter-level relationship R does not require any gating mechanism. The computation of A and O can be implemented using either LA or GLA, whereas GSA requires two GLA operations. Since the additional LA computation and the sequential multiplication with A constitute a relatively small portion after fusing them into a single operator, the overhead is much smaller than that of GLA. Consequently, the overall cost of RSA is approximately 1.4× that of GSA with the same state size. We also observe that replacing the LA operation for computing R with a simple learnable matrix results in only a modest performance drop, while still outperforming GSA.
>
> **Inference times (ms)**
>
> |Seq Len|FlashAttention|RSA-4 x 32|RSA-4 x 16|GSA-512|GSA-256|
> |-|-|-|-|-|-|
> |16K|117.51|156.69|80.03|245.77|159.76|
> |32K|331.92|306.67|157.06|487.49|315.98|
> |64K|1057.79|608.23|320.67|977.67|633.21|
> |128K|3801.54|1218.08|632.03|1961.19|1278.01|
> |256K|14420.34|2439.89|1273.23|3922.17|2541.24|
>
> |Model|LMB |PIQA |Hella. |Wino. |ARC-e |ARC-c |SIQA |BoolQ |Avg |mmlu|
> |-|-|-|-|-|-|-|-|-|-|-|
> |RSA-4 x 32|32.45|66.52|39.04|51.68|56.80|27.85|37.23|60.09|46.45|25.61|
> |GSA-256|30.82|66.40|38.10|50.56|55.56|26.19|37.20|60.01|45.61|23.07|
>
>
> # Q2
> Refer W3
>
> **Details: https://anonymous.4open.science/r/icml26-rsa-29B9/**

---

### Official Review · Reviewer_39Kc · 2026-03-13

**Soundness:** 3
**Presentation:** 2
**Significance:** 4
**Originality:** 3
**Overall Recommendation:** 5
**Confidence:** 3

**Summary:**

The paper proposes Recursive Sparse Attention (RSA), a linear-time sequence modeling mechanism intended to close the gap between compressed-state linear models and full attention. The method combines two components: a hierarchical deep–shallow memory that stores coarse information in shallow states and residual information in deeper states, and a correlation-aware sparse readout that uses inter-level relations plus thresholded activation to retrieve only salient memory components. The paper argues that this design better preserves information capacity and reduces attention diffusion, and evaluates RSA in autoregressive language modeling, retrieval, passkey-style long-context recall, and LongBench-style long-sequence tasks. Empirically, RSA is usually stronger than several linear-attention baselines and often competitive with Transformer baselines, though it does not uniformly surpass strong pretrained Transformers in the fine-tuning setting.

**Compliance With Llm Reviewing Policy:**

Affirmed.

**Final Justification:**

This is my final score.

**Key Questions For Authors:**

Why does RSA still lag the pretrained Qwen baseline on the strongest fine-tuning comparison? RSA-7B-Qwen remains below Qwen2.5-7B average and on MMLU/overall LongBench average. Is this due to optimization instability, architectural bottlenecks, insufficient fine-tuning budget, or irrecoverable loss from replacing dense attention?

**Limitations:**

yes

**Strengths And Weaknesses:**

Soundness

The paper has a clear technical agenda and the proposed mechanism is nontrivial. The decomposition into hierarchical storage plus sparse readout is coherent, and the sequential formulation is explicit enough to understand the intended update/readout pathway. The experimental section is reasonably broad for a systems-style architecture paper: it includes training-from-scratch results at 360M and 1.3B scale, fine-tuning experiments on Qwen backbones, retrieval benchmarks, passkey recall, LongBench, efficiency measurements, and ablations. In particular, RSA does appear stronger than several compressed-state baselines on retrieval-style tasks and maintains perfect passkey accuracy at 32K in the reported setup.

That said, the soundness story is incomplete in several important respects. First, the key theoretical claims are stated at a high level and deferred to the appendix, while the main text offers only limited discussion of assumptions, approximation error, and what exactly “matching the storage capacity of standard attention” means operationally. Theorems 1 and 2 are suggestive, but as presented in the main paper they do not yet establish equivalence in function class, optimization behavior, or finite-precision generalization. Second, many comparisons mix reported numbers from prior work with the authors’ own runs, which weakens strict apples-to-apples interpretation. Third, the strongest fine-tuning comparison is mixed: RSA improves substantially over GSA, but still trails the original pretrained Qwen-2.5-7B on average and on MMLU. Fourth, the efficiency claim is not wholly favorable: RSA is reported as about 1.4× slower than GSA at equivalent total state capacity, which is a meaningful cost for an already efficiency-motivated line of work. Finally, LongBench results are positive against linear baselines, but RSA-7B-Qwen remains below the original Qwen2.5-7B average, which limits the claim that the method approaches or surpasses full attention in a broad sense.

Presentation

The paper is organized in a conventional and mostly readable manner. Unfortunately, Figure 1 is messy and do not help to understand picture of storage, state relations, and sparse readout very much. The method section makes the intended intuition reasonably clear: coarse-to-fine memory, residual refinement, and selective retrieval.

The writing is often overextended and mixes technical claims with speculative neuroscience analogies and interpretability commentary that do not materially strengthen the paper. Several core definitions are harder to parse than necessary, notation is somewhat dense, and the transition from intuitive arguments to formal claims is loose. In particular, the paper should be clearer about what is novel relative to recent multi-state / delta-rule / compressed-KV lines such as MVA, MoM, PLA, and GSA, because the proposal currently reads as a substantial synthesis rather than a sharply isolated new principle.

Significance

The problem is important. Efficient long-context alternatives to full attention remain relevant, and a method that materially improves retrieval and long-range reasoning without reverting to dense attention would be useful. RSA appears meaningfully stronger than several linear baselines on retrieval benchmarks and passkey recall, and the 1.3B from-scratch retrieval average is close to Transformer++ while exceeding most non-Transformer baselines reported in the table. That is a substantive positive result.

The limitation is that the practical significance is moderated by the compute/performance tradeoff and by the fact that strong pretrained Transformers still remain stronger in the fine-tuned regime on the most realistic comparisons shown. Thus, the paper may be significant for the subcommunity working on efficient attention and recurrent/linear alternatives, but the evidence here does not yet show a broadly superior replacement for standard attention.

Originality

The work is original at the level of synthesis and architecture design: hierarchical residual memory plus explicit inter-level relation matrices plus sparse readout is a nontrivial combination, and the emphasis on readout rather than only state update is a useful perspective. That is the most convincing aspect of the paper’s novelty claim.

At the same time, the paper is not radically novel in ingredients. It builds heavily on existing themes: multi-state memory, delta-style updates, KV compression, sparse/dynamic routing. The paper’s originality therefore lies more in integration and framing than in introducing a clearly isolated primitive. I would score originality as moderate rather than high.

Overall, my assessment: promising and technically interesting, with credible empirical strengths on retrieval-oriented long-context tasks, but not yet fully convincing in its broader claim of approaching or surpassing full attention.

---

> ### Author Rebuttal · Authors · 2026-03-30
>
> **Key Questions:**
> Thank you very much for your valuable feedback. We believe the primary reasons are the quality of the fine-tuning data and an insufficient fine-tuning budget.
>
> First, we used public datasets. Since large language models are typically released with supervised fine-tuning (SFT) on carefully curated data, fine-tuning them on public datasets can lead to a certain degree of performance degradation. Consequently, fine-tuning such models with relatively lower-quality data can reduce their performance. To illustrate this, we fine-tuned Qwen2.5-7B-CPT using the same fine-tuning data as RSA-7B:
>
> |Model|Tokens|PIQA|Hella.|Wino.|ARC-e|ARC-c|SCIQ|MMLU|Avg|
> |-|-|-|-|-|-|-|-|-|-|
> |Qwen2.5-3B*|18.0T|78.6|73.5|68.5|77.4|45.0|96.2|65.7|72.13|
> |Qwen2.5-3B-CPT|20B|76.1|72.0|68.6|77.2|45.9|95.2|55.6|70.09|
> |RSA-3B-Qwen (Ours)|100B|79.8|75.1|72.4|79.6|46.5|96.2|55.8|72.20|
> |Qwen2.5-7B*|18.0T|78.8|79.0|73.1|80.4|51.4|96.6|74.2|76.21|
> |Qwen2.5-7B-CPT|10B|77.5|77.9|72.3|80.0|52.4|96.2|63.5|74.26|
> |Qwen2.5-7B-CPT|20B|77.1|77.4|72.1|80.3|52.1|96.3|61.9|73.91|
> |RSA-7B-Qwen (Ours)|20B|79.6|80.1|73.8|82.1|52.6|96.2|59.1|74.79|
>
> Second, the number of tokens used for fine-tuning is insufficient. The Qwen series, are pre-trained on up to 18.0T tokens. Therefore, fine-tuning with only 20B tokens (approximately 0.1% of the pre-training data) may not be sufficient to fully convert the distribution modeled by standard attention to that of RSA.
> |Model|Tokens|MMLU|SQA|MQA|Sum|FS|Code|LongBench Avg.|
> |-|-|-|-|-|-|-|-|-|
> |Qwen2.5-7B|18.0T|74.2|28.60|28.71|17.85|69.51|58.60|40.65|
> |Qwen2.5-7B-CPT|20B|61.1|27.91|28.12|17.64|69.52|58.00|40.24|
> |RSA-7B|20B|59.1|25.25|23.68|16.96|68.96|58.00|38.57|
> |RSA-7B|30B|60.2|26.49|24.52|17.05|69.56|59.00|39.33|
>
> Future improvements could include incorporating high-quality datasets or using distillation.
>
> Finally, we would like to explain why we believe RSA establishes a solid foundation for linear models that can match or even surpass standard attention. We are confident that a reviewer of your expertise can recognize that the improvements introduced by RSA have the potential to exceed standard attention, and we hope this will further strengthen the evaluation of our work.
>
> Our reason:
>
> We argue that linear models incorporate a valuable inductive bias compared to the in-context learning of standard attention. Specifically, they can compress all learned data into RSA’s memory during training. Given that RSA’s memory capacity is sufficiently large (Based on the proposition in the PLA paper, it is easy to infer that when the number of levels equals the head dimension, the capacity approaches infinity.), this enables large language models to accumulate experiences and evolve over time—similar to human learning—which is a significant advantage of linear attention.
>
> A concrete analogy: the learning process of the standard Transformer lacks an inherent temporal inductive bias, the accumulation of knowledge over time. It can be compared to a human with strong intelligence but no memory or experience. While the MLP layers store a certain amount of knowledge, this knowledge is heavily compressed and tends to be activated only within fixed contexts (liken MLP to human genetics). **Standard attention functions like a specialist who, despite rigorous training, experiences amnesia, which greatly limits its capabilities. This is why methods like in-context learning or few-shot prompting are needed to enhance its performance by adding temporary memory.**
>
> Our model lays a strong foundation for overcoming this limitation. First, the memory capacity of our model increases exponentially, and its read mechanism is highly effective. Such a model support a new form of modeling and learning that is unattainable by standard Transformers or existing linear models. Specifically, it supports incremental memory learning, which simulates the process of human growth by maintaining a fixed-size state with sufficient capacity (unlike other linear models, which suffer from insufficient capacity). During learning, knowledge is continuously integrated into this state (the "brain"), a capability that standard attention lacks. Standard attention can only extend the KV cache indefinitely, and even adding fixed-size learnable parameters is not viable, as its read mechanism is too simplistic to effectively organize and retrieve memories. Other linear models, while possessing a fixed-size state, cannot match our performance because our memory capacity and reading ability are exponentially stronger than these traditional approaches.
>
> **In summary, standard attention acts as a memoryless, amnesiac expert, whereas the future lies in models that integrate sustained memory with expressive read mechanisms. Our RSA framework establishes a solid foundation and realizes this through a high-capacity state and a robust retrieval architecture.**
>
> **weakness and all details: https://anonymous.4open.science/r/icml26-rsa-29B9/**

---

> > ### Author Rebuttal · Reviewer_39Kc · 2026-04-03
> >
> > I have no more questions to the authors. I believe the paper is not ideal but worth to be presented on the conference.

---

> > > ### Author Response · Authors · 2026-04-03
> > >
> > > **Thank you for your recognition and support. It is also a great honor to once again contribute to the ICML community.**
> > >
> > > We will continue to develop and promote our method, exploring a meaningful path toward the next-generation architecture and training paradigm. **We believe that the combination of three elements—theoretically exponential or infinite storage capacity, increasingly powerful read capability, and a training paradigm that continuously absorbs temporal sequence memory into the state during training to form intelligent experiential knowledge and memory—represents one of the future directions for achieving human-like intelligence.** We have already made certain experimental progress, and we hope that this approach will evolve into a mainstream pathway toward genuine intelligence and, in doing so, further benefit the ICML community.
> > >
> > > **Finally, we thank you once again for your valuable comments and suggestions, as well as the efforts of all the reviewers and the Area Chair.** Thank you for giving us the courage and determination to continue on this path.

---

### Official Review · Reviewer_CQjr · 2026-03-17

**Soundness:** 3
**Presentation:** 2
**Significance:** 3
**Originality:** 3
**Overall Recommendation:** 3
**Confidence:** 3

**Summary:**

The paper describes a novel alternative to self-attention -- recursive sparse attention (RSA) -- that aims to keep the benefits of linear attention (linear complexity with sequence length, bounded state size), while enhancing the expressive power through greater selectivity via recursive hierarchical processing and sparse activation. The paper is motivated by human memory, where memories are often recalled first at a course level before being explored at finer grained levels.
The authors provide a lengthy motivation for their work based on prior linear attention approaches, including more recent gated, slot-based, and delta-rule based variants, a discussion of human memory, and two theorems. The first theorem describes the number of token types that are representable as the product of the dimensions of different states. The second theorem describes a recursive formulation of multi-key state systems, where higher-level keys can be recovered from an MLP over lower level keys and a self-attention over those lower level keys with a key state vector. They then describe the recurrent inference formulation of their approach, along with an approximate parallel variant. A sparse readout mechanism is described.
The authors perform an empirical analysis of their method using language tasks, in both from-scratch and fine-tuning settings. In fine-tuning, RSA generally performs favorably to base models of comparable size, even outperforming the base Qwen 3B model from which it was fine-tuned when fine-tuned with 100B tokens (c.f. 18T token pre-training). Notably, RSA vastly outperforms gated slot attention (GSA) which is a related linear attention method. The 7B variants also perform well, albeit not outperforming the base Qwen 7B model.
In the fine-tuning section, at both 360M and 1.3B sizes, RSA outperforms all other linear attention baselines, and performs close to but worse than the base Transformer++ model; RSA is the only model trained by the authors, whereas the other baselines are extracted from results reported in the literature.
The authors go on to explore the long sequence skills of the model. They find that, similar to base transformers, the models can perform retrieval perfectly over very long sequences, with no degradation up to 32K tokens even when trained on 8K tokens. In contrast, GSA and GLA significantly degrade.
Last, the authors analyze the efficiency of their model in comparison to GSA. They find that it is about 1.4x slower , and is moderately more memory intensive.
The model appears to be a drop-in replacement for self-attention, with many of the benefits of linear attention methods, with stronger performance and vastly superior long-sequence retrieval abilities.

**Compliance With Llm Reviewing Policy:**

Affirmed.

**Key Questions For Authors:**

- how does RSA compare in efficiency relative to the base model and other linear attention variants?
- how many additional trainable parameters does RSA introduce, for each of the variants? does the additional parameter count meaningfully nudge the total parameter count of the model?
- how good - empirically - is the parallel approximation to the recurrent formulation?
- does theorem 2 add anything beyond the universal approximation theorem?
- what human data actually motivates the approach? this motivation is very vague, and some references would be helpful (general comment - paper is very thin in references).

**Limitations:**

Yes

**Strengths And Weaknesses:**

Soundness: overall, the paper is mixed in terms of technical soundness. The mathematical notation is sloppy at times, for example, there are several cases in which the matrix diagonalization of a vector must be inferred.

Presentation: Theorem 2 is not really a theorem so much as a reframing of the universal approximation theorem, used to motivate their particular architecture. Overall, too much time is spent motivating and introducing the technique, at the expense of stronger and more detailed experimental results. This is particularly noteworthy because the theoretical justification is merely justification; as far as I can tell, it does not rigorously prove anything. The experimental results are impressive but limited. Notably, linear attention methods are typically selected for their efficiency benefits. This paper does not present any efficiency benefits of their model. The only efficiency results presented highlight a decrement relative to GSA; this latter result is not inherently problematic, because RSA outperforms GSA by a large margin, but it is noteworthy that their are no positive efficiency results for RSA. For less familiar readers, this is an unfortunate gap that could prevent wider adoption. Last, despite the extensive amount of space dedicated to the mathematical description, I found it hard to follow due to the many superscripts, notational inconsistencies, and excessive substitutions.

Significance: the paper has the potential to be very significant. I found the architecture well motivated and the results impressive. I would like to see analyses across longer context lengths. Additionally, I would like to see demonstrations that the improvements are not merely due to the fine-tuning. For example, in comparing to the base model, it is surprising to see the RSA variant outperform it after finetuning. This suggests that the finetuning dataset may be doing a lot of the heavy lifting. I would also like to see extensive analyses of the efficiency of the model. This is much more important than showing weak from-scratch training results

Originality: I think the paper is original, and has a nice connection to human memory that could be expanded upon in future work.

---

> ### Author Rebuttal · Authors · 2026-03-30
>
> First of all, thank you very much for your valuable feedback. Below, we provide a response to your question. As for the detailed response and weaknesses: **https://anonymous.4open.science/r/icml26-rsa-29B9/**.
>
> # Q1
>
> We believe efficiency comparisons are most meaningful when performance is comparable. To this end, we trained RSA with 4 levels and state sizes of 32, and compared against GSA with a state size of 256. Our method achieves slightly better performance with substantially lower computation, demonstrating higher efficiency.  All configurations use a 16-layer decoder architecture of LA + MLP.
>
> **Inference times (ms)**
>
> |Seq Len|FlashAttention|RSA-4 x 64|RSA-4 x 32|RSA-4 x 16|GSA-512|GSA-256|GLA-512|GLA-256|GDN-256|
> |-|-|-|-|-|-|-|-|-|-|
> |8K|47.69|103.55|82.01|41.73|125.16|82.07|102.65|69.01|68.34|
> |16K|117.51|199.96|156.69|80.03|245.77|159.76|201.79|134.24|134.56|
> |32K|331.92|392.32|306.67|157.06|487.49|315.98|400.73|265.46|268.08|
> |64K|1057.79|780.40|608.23|320.67|977.67|633.21|801.59|532.05|521.02|
> |128K|3801.54|1568.54|1218.08|632.03|1961.19|1278.01|1607.77|1066.78|ERR|
> |256K|14420.34|3137.63|2439.89|1273.23|3922.17|2541.24|3217.24|2130.93|ERR|
>
> |Model|LMB |PIQA |Hella. |Wino. |ARC-e |ARC-c |SIQA |BoolQ |Avg |mmlu|
> |-|-|-|-|-|-|-|-|-|-|-|
> |RSA-4 x 32|32.45|66.52|39.04|51.68|56.80|27.85|37.23|60.09|46.45|25.61|
> |GSA-256|30.82|66.40|38.10|50.56|55.56|26.19|37.20|60.01|45.61|23.07|
>
> Our method provides an efficient means of scaling state capacity. The effective capacity grows exponentially with the number of levels: in our configuration, capacity is 64^4, whereas GSA uses a capacity of 256. To avoid confusion, we will revise “capacity” to “size” in line 425 of the paper. A truly fair efficiency comparison would require GSA to match this capacity.
> More importantly, scaling the state size of GSA, GLA, or DeltaNet still renders certain tasks—such as very long passkey retrieval or complex MMLU benchmarks—infeasible. Our model handles these tasks effectively, which we consider a key advantage.
>
> We conclude with a broader perspective. Current linear attention research spans multiple directions, where efficiency is an important but not exclusive criterion. Other critical axes include scalability and performance. Our work provides evidence that linear models can scale effectively and achieve competitive results. More importantly, linear attention offers advantages beyond efficiency—specifically, it enables forms of modeling and learning that are challenging for standard attention. We view this as the key direction to pursue; otherwise, linear attention risks remaining a niche solution confined to resource-constrained settings. For more details, please see Reviewer 39Kc's our reply.
>
> # Q2
>
> RSA introduces very few additional parameters, accounting for less than 0.05% of the total. Specifically, the gating mechanism introduces 4 x state_len x num_heads x head_dim = 0.25M and  MLP introduces 4 x (2 x (2x head_dim) x head\_dim) = 0.25M parameters, which are shared for K and V. For a 7B model such as Qwen2.5-7B (with 7,615.616512M parameters), this additional parameter count is negligible—less than 0.05%.
> For models trained from scratch, we compensate for the parameter difference by using fewer KV heads compared to baseline models.
>
> # Q3
>
> The approximation error is very small. In fact, replacing the query-state process in the parallel formulation, $\bar{f}^{(i)}(\cdot) = \bar{f}^{(i)}\left(k_t^{(i)} {S_{t-1}^{K(i)}}^{\top}\right)$, with EQ1: $\bar{f}^{(i)}(\cdot) = \bar{f}^{(i)}\left(k_t^{(i)} {W_C^{K(i)}}^{\top}\right)$—where $W_C$ is the learnable gating parameter mentioned above—yields comparable performance.
>
> We evaluated the model under three settings: using the parallel mode, using the recurrent mode, and using EQ1 for training and testing. The results show virtually no difference in performance.
>
> |Model|Wiki ppl↓|LMB ppl↓|LMB|PIQA|Hella.|Wino.|ARC-e|ARC-c|SIQA|BoolQ|Avg|
> |-|-|-|-|-|-|-|-|-|-|-|-|
> |RSA-360M test: recurrent|27.05|35.86|32.89|66.50|39.24|52.04|56.70|27.81|37.24|60.12|46.57|
> |RSA-360M test: parallel|27.01|35.89|32.80|66.54|39.20|52.04|56.72|27.86|37.22|60.12|46.56|
> |RSA-360M Wc|27.06|35.82|32.91|66.47|39.24|52.01|56.81|27.78|37.24|60.09|46.57|
>
> # Q4
> Theorem 2 is indeed grounded in the universal approximation theorem. However, it provides an intuitive and interpretable perspective.
>
> # Q5
> Our approach was partially inspired by general knowledge from the neuroscience of learning, along with a few references. However, the primary motivation came from our own introspection on how information is processed and learned. As a human, I modeled this process based on my own cognitive experience, which we believe constitutes a brain-inspired approach. We agree that incorporating more references would strengthen the paper, and if you have any suggestions for relevant literature, we would greatly appreciate them and would be happy to include them.

---

> > ### Author Rebuttal · Reviewer_CQjr · 2026-04-04
> >
> > Thanks for the replies. I am open to shifting my score to reflect recommending acceptance but I have a few unresolved questions.
> >
> > Resolved: Q2, Q3, Q4
> >
> > Unresolved:
> > Q1:
> > - from your table, I do not see your point that RSA-4x32 achieves "substantially lower computation" than GSA-256. Granted, the performance is slightly better as stated, but the inference time seems basically identical.
> > - "More importantly, linear attention offers advantages beyond efficiency—specifically, it enables forms of modeling and learning that are challenging for standard attention. We view this as the key direction to pursue". I'm not sure I follow this point, mind elaborating and stating clearly how the paper demonstrates these advantages for the proposed algorithm in particular?
> >
> > Q5:
> > - As a general comment, It is a bit insulting to the academics who have devoted their life to studying human memory that you can claim inspiration about human memory and not spend a few hours reading up about what is actually known through decades of study to back up your claims. You say "As a human, I modeled this process based on my own cognitive experience, which we believe constitutes a brain-inspired approach." Historically, introspection has proven a quite limited method for studying human cognition -- this is what led to the popularization of behaviorism -- though its a reasonable first start. Please either review the literature on human memory and incorporate some relevant citations to support your claims of brain inspiration, or do not claim brain inspiration. It is your job as the author to read broadly enough to justify your claims with solid references. This is how science percolate into new spaces.

---

> > > ### Author Response · Authors · 2026-04-04
> > >
> > > Thank you very much for your response and valuable comments. Below I provide detailed replies to your questions, hoping to resolve your concerns.
> > >
> > > **Unresolved: Q1:**
> > >
> > > 1. Regarding this issue, we admit that our description was problematic, and we sincerely apologize for that. We will change "substantially lower computation" to "slightly lower computation". The reason is that we originally intended to train both RSA-4×32 and RSA-4×16 to demonstrate that our efficiency is better than GSA-256, which is why we included RSA-4×16 in the table. However, due to time and resource constraints at the time, RSA-4×16 had not yet finished training, so the word "substantially" was not removed in time. Thank you very much for pointing this out. RSA-4×16 has now been completed, and we present its performance comparison below. RSA-4×16 is indeed more efficient than GSA-256.
> > >
> > > |Model|LMB|PIQA|Hella.|Wino.|ARC-e|ARC-c|SIQA|BoolQ|Avg|mmlu|
> > > |-|-|-|-|-|-|-|-|-|-|-|
> > > |RSA-4 x 16|30.96|66.28|38.49|50.98|56.05|28.03|37.16|59.98|45.99|25.28|
> > > |GSA-256|30.82|66.40|38.10|50.56|55.56|26.19|37.20|60.01|45.61|23.07|
> > >
> > > 2. Let us explain why linear models enable forms of modeling and learning that are challenging for standard attention (also be found in Reviewer 39Kc). Specifically:
> > >
> > > Here is the simplified version and Detail in link https://anonymous.4open.science/r/icml26-rsa-29B9/:
> > >
> > > First, linear models maintain a fixed-size state for sequence memory, while standard attention uses an unbounded KV cache. This lets linear models explicitly learn and memorize: during training, temporal tokens compress into a fixed state, like storing life experiences in our brain—something standard attention cannot do. Both also have implicit memory in FFN layers. Explicit memory is crucial; that's why standard attention needs in-context learning or few-shot to inject prior knowledge. Thus, after learning, RSA becomes an expert with genuine memory, whereas standard attention is an intelligent but amnesiac expert—though their neural pathways are established, requiring extra context to perform.
> > >
> > > Although linear models could adopt this paradigm, most lack the potential to fully realize this kind of modeling, because we believe that sufficient state capacity and a powerful read mechanism capable of precisely locating relevant content are required.
> > >
> > > (1) Most existing linear models have effective capacity only at or a few times their state size. Ours achieves exponential capacity. Moreover, PLA showed standard attention is a special case of linear models with bounded state size—so an unbounded sequence can be compressed into a fixed state. They didn't prove how many levels are needed. In a future revision, we will add that head-dimension state size suffices for near-infinite capacity. This doesn't affect our current RSA results; exponential capacity is already large enough.
> > >
> > > (2) MoM, MVA, and PLA explored state expansion; They have near-exponential capacity but poor read mechanisms: MVA focuses on short sequences, PLA reads all memories at once (noisy). RSA's Correlation-Aware sparse read has potential, though further read improvements may be needed.
> > >
> > > The paper does not currently prove this part of the algorithm; we will revise the future work section to address this, as this direction is complex enough to constitute a substantial standalone contribution. We aim for a Nature-level journal; even if not accepted, we believe in this potential. We will also improve FFN components via analogy to human DNA, moving toward human-like modeling and architecture.
> > >
> > > **Q5:**
> > >
> > > We sincerely apologize that this part made you uncomfortable. Following your advice, we will adopt both suggested improvements.
> > >
> > > First, we will introduce a substantial number of references on human memory, including but not limited to the following:
> > >
> > > [0] Human memory: A proposed system and its control processes.
> > > [1] Cognitive neuroscience perspective on memory: ...
> > > [2] Integrating spatial working memory and remote memory: ...
> > > [3] Brain-inspired computing systems: a systematic literature review.
> > > [4] Conscious and unconscious memory systems.
> > > [5] Contemporary neurocognitive models of memory: ...
> > > [6] Episodic memory formation: A review of complex hippocampus input pathways.
> > > [7] Brain-inspired computing systems: A systematic ...
> > >
> > > Since we are not researchers in that field, if there are any important references we have missed, please let us know and we will be happy to cite them.
> > >
> > > Second, we will also revise the wording in the paper regarding "brain inspiration" and "neuroscience-inspired," changing it to phrases such as "based on our observations and intuition." These changes will appear in no more than four places.
> > >
> > > If you believe further modifications are needed, please let us know.
> > >
> > > Finally, thank you very much for your comments and feedback. We look forward to your reassessment and further discussion. Thank you.

---

### Decision · Program_Chairs · 2026-04-30

**Decision:**

Reject

**Comment:**

The paper received a combination of positive and negative reviews. It proposes Recursive Sparse Attention, which combines benefits of linear models and standard attention. While several reviewers see potentially interesting ideas described in this submission, most agree that the paper is poorly written and very hard to understand. This poor presentation creates a problem for evaluating the validity of the core technical content.

A discussion between the AC and Reviewers has been initiated. As a result of this discussion, even reviewers who voiced a strong support for this work eventually arrived at the conclusion that the mechanism behind Recursive Sparse Attention, proposed in this work, is unclear from the presentation. For this reason, the paper cannot be accepted in its current form. It needs a substantial revision to address this issue.

Despite this, all reviewers agree that the paper tackles an important problem. I encourage the authors to consider revising the presentation and resubmitting to a future venue.